# Summaries as Centroids for Interpretable and Scalable Text Clustering

**Jairo Diaz-Rodriguez**
Department of Mathematics and Statistics
York University
Toronto, Ontario M3J 1P3 jdiazrod@yorku.ca

## Abstract

We introduce k-NLPmeans and k-LLMmeans, text-clustering variants of k-means that periodically replace numeric centroids with textual summaries. The key idea, *summary-as-centroid*, retains k-means assignments in embedding space while producing human-readable, auditable cluster prototypes. The method is LLM-optional: k-NLPmeans uses lightweight, deterministic summarizers, enabling offline, low-cost, and stable operation; k-LLMmeans is a drop-in upgrade that uses an LLM for summaries under a fixed per-iteration budget whose cost does not grow with dataset size. We also present a mini-batch extension for real-time clustering of streaming text. Across diverse datasets, embedding models, and summarization strategies, our approach consistently outperforms classical baselines and approaches the accuracy of recent LLM-based clustering without extensive LLM calls. Finally, we provide a case study on sequential text streams and release a StackExchange-derived benchmark for evaluating streaming text clustering.

## 1 Introduction

Text clustering is a core problem in natural language processing (NLP), with applications in document organization, topic exploration, and information retrieval (Schütze et al., 2008; Steinbach, 2000). A standard pipeline embeds documents into vectors (Devlin, 2018; Sanh, 2019; Mikolov, 2013; Pennington et al., 2014; Brown et al., 2020; Jin et al., 2023) and then groups them with a clustering algorithm (Petukhova et al., 2025). Among these algorithms, k-means (MacQueen, 1967) remains ubiquitous, iteratively updating each centroid as the mean of its assigned points. While effective, this purely numerical averaging can blur contextual nuance present in the original texts (Reimers & Gurevych, 2019). Prior work has explored alternative centroid definitions and related objectives (Jain & Dubes, 1988; Bradley et al., 1996; Kaufman & Rousseeuw, 2008), yet these approaches remain anchored in vector space, which limits interpretability and can induce semantic drift between centroids and their underlying documents. This motivates centroids that are explicitly textual, aligning prototypes with human-interpretable summaries.

**Our proposal.** We introduce a simple modification to k-means: periodically replace numerical centroid updates with summarization steps. Rather than averaging embeddings at every iteration, we compute, at spaced iterations, a textual prototype that summarizes each cluster and then re-embed it with the same encoder to obtain the centroid used for assignments. These *summary-as-centroid* updates remain inside the standard k-means loop but capture richer contextual meaning. Alternating numerical and summary-based updates yields clusters that are more interpretable and often more semantically coherent. Figure 1 illustrates our proposal and shows how even a single summarization step can redirect k-means toward a qualitatively improved solution.

Our goal is not to strictly dominate every clustering algorithm, but to provide a simple novel approach that (i) improves clustering quality over standard k-means–style methods, (ii) yields human-readable centroids, and (iii) scales to large and streaming datasets with limited LLM usage.

**Summarization step.** We consider two families of summarizers: (1) Classical NLP such as centroid-based summarization, graph-based methods such as TextRank , and LSA-style techniques

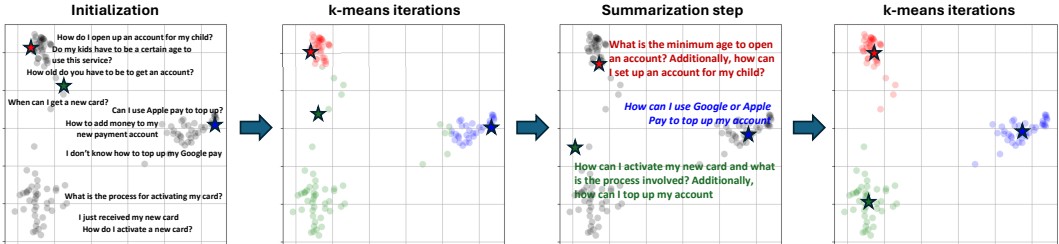

Figure 1: Illustration of k-NLPmeans/k-LLMmeans with a single summarization step. First panel shows the text embeddings with stars marking the initial centroids; second shows the partition reached after k-means iterations (a local minimum); third performs the summarization step, each previous cluster is summarized into a textual prototype and re-embedded; final panel runs one more k-means iteration using these summaries as centroids, yielding a qualitatively improved partition. This example is illustrative; in practice, clusters can be multi-topic, and summaries may describe multiple prominent subtopics for a single cluster.

(Radev et al., 2004; Mihalcea & Tarau, 2004; Deerwester et al., 1990), which are fast and deterministic (yielding k-NLPmeans); and (2) LLM-based summaries that can capture richer context and filter secondary or noisy content (Zhang et al., 2020a; Raffel et al., 2020; Jia & Diaz-Rodriguez, 2025) (yielding k-LLMmeans). In both variants, summaries are recomputed only at scheduled iterations, so the algorithms share the same *summary-as-centroid* mechanism and differ solely in the summarizer. While methods from (1) and (2) are standard in summarization, our periodic use of summary-derived centroids within the k-means loop is, to our knowledge, novel and central to the method's interpretability.

**Interpretability and scalability.** Each summarization step produces a concise, human-readable centroid that exposes evolving cluster semantics, simplifying debugging, validation, and labeling. The summarizer is modular: it can be LLM-free (k-NLPmeans) or LLM-assisted under a fixed per-iteration budget that does not grow with dataset size (k-LLMmeans). For large-scale and streaming settings, we adapt mini-batch k-means (Sculley, 2010) by inserting summarization steps into the mini-batch update rule, enabling efficient online clustering while retaining interpretability.

**Relation to existing LLM-based clustering.** Recent work shows that LLMs can deliver strong unsupervised clustering (Zhang et al., 2023; Feng et al., 2024; De Raedt et al., 2023; Viswanathan et al., 2024; Shi & Sakai, 2023; Tarekegn et al., 2024; Nakshatri et al., 2023). However, many pipelines face two practical issues: (i) scalability, as they may rely on semi-supervision, iterative labeling, or a number of LLM calls that grows with dataset size; and (ii) opaque optimization, since they often combine prompts, greedy merges, and similarity thresholds without an explicit objective, making convergence behavior hard to analyze. Our k-LLMmeans addresses both: by summarizing per cluster at spaced iterations, we cap LLM usage independently of dataset size, and by keeping assignments and numeric updates in embedding space, we preserve the standard k-means objective between summary steps. In both k-NLPmeans and k-LLMmeans, if the summaries are weak, the procedure gracefully degrades to vanilla k-means with a suboptimal initialization, retaining its standard local-convergence behavior. Note that our method is LLM-optional: k-NLPmeans replaces the summarizer with classical NLP techniques, incurring no LLM usage.

**Contributions.** In summary, we (i) propose k-NLPmeans and k-LLMmeans, a *summary-as-centroid* variant of k-means that periodically replaces numeric centroids with textual prototypes re-embedded by the same encoder, an LLM-optional design that, by construction, preserves the standard k-means objective between summary steps; interpretability follows from human-readable centroids and transparent intermediate outputs; (ii) extend the approach to mini-batch k-means for efficient, online clustering of streams; (iii) present a comprehensive empirical study across datasets, embeddings, and summarizers, showing consistent gains over classical baselines and competitiveness with recent LLM-based clustering under a dataset-size–independent LLM budget; and (iv) provide a case study on sequential text streams and release a StackExchange-derived benchmark for evaluating streaming text clustering.

## 2 PRELIMINARIES: K-MEANS FOR TEXT CLUSTERING

Given a corpus of $n$ text documents $D = \{d_1, \cdots d_n\}$. Each document $d_i$ is represented as a $d$-dimensional embedding vector $\mathbf{x}_i \in \mathbb{R}^d$ such that:

$$\mathbf{x}_i = \text{Embedding}(d_i).$$

The goal of k-means clustering is to partition these $n$ document embeddings into $k$ clusters, minimizing the intra-cluster variance. Formally, we define the clustering objective as:

$$\min_{C_1, C_2, \ldots, C_k} \sum_{j=1}^{k} \sum_{i \in [C_j]} \|\mathbf{x}_i - \boldsymbol{\mu}_j\|^2, \tag{1}$$

where $C_j$ denotes the set of embeddings assigned to cluster $j$, $[C_j] = \{i | \mathbf{x}_i \in C_j\}$ denotes the set of embedding indices assigned to cluster $j$ and $\boldsymbol{\mu}_j$ is the cluster centroid, computed as the mean of the assigned embeddings:

$$\boldsymbol{\mu}_j = \frac{1}{|C_j|} \sum_{i \in [C_j]} \mathbf{x}_i. \tag{2}$$

Lloyd's algorithm (Lloyd, 1982), the standard heuristic for k-means, alternates between assigning each document embedding $\mathbf{x}_i$ to its nearest centroid and recomputing each centroid as the mean of its assigned points. Repeating these two steps for $T$ iterations monotonically reduces the within-cluster sum of squared distances, steering the procedure toward a (locally) optimal set of centroids. However, due to its sensitivity to initialization and the non-convex nature of its objective function, k-means does not guarantee convergence to the global optimum and can instead become trapped in local optima (MacQueen, 1967; Lloyd, 1982). Various strategies, such as k-means++ initialization and multiple restarts, have been proposed to mitigate these issues and improve the likelihood of achieving better clustering results (Arthur & Vassilvitskii, 2006).

## 3 K-MEANS WITH SUMMARIZATION STEPS

We enhance k-means for text clustering by periodically replacing the numerical centroid update with *summarization steps* that yield textual-summary–based centroids. The procedure (Algorithm 1 in Appendix D) is identical to k-means algorithm except that every $l$ iterations the mean update in equation 2 is replaced by the embedding of a cluster textual summary. At all other iterations, the standard update is used (see Figure 1 for an illustration with one summarization step). The summarizer can be instantiated with classical, deterministic methods (k-NLPmeans) or with LLM-based summaries (k-LLMmeans), and is not restricted to the specific techniques we discuss, any textual summarization operator can be dropped in.

### 3.1 K-NLPMEANS.

Our first variant uses classical extractive summarization to compute a textual prototype in place of the standard centroid update. Formally, we replace equation 2 with

$$\boldsymbol{\mu}_j = \text{Embedding}\left(f_{\text{NLP}}^{(q)}(S_j)\right), \tag{3}$$

where $S_j$ is the collection (multiset) of sentences obtained by tokenizing all documents assigned to cluster $j$. Unless otherwise noted, sentence–sentence similarities are computed as cosine similarity between sentence embeddings produced by the *same* encoder used for documents. The operator $f_{\text{NLP}}^{(q)}$ returns a short summary of $q$ sentences; typical instantiations include:

- *Centroid-based summarization* (Radev et al., 2004): compute the centroid of the sentence embeddings for $S_j$; rank sentences by cosine similarity to this centroid; concatenate the top $q$ sentences (by rank) to form $f_{\text{NLP}}^{(q)}(S_j)$.
- *Graph-based (TextRank)* (Mihalcea & Tarau, 2004): build a graph whose nodes are sentences in $S_j$ with edge weights given by pairwise cosine similarity of their embeddings; run a PageRank-style algorithm to score sentences; select and concatenate the top $q$.

- *LSA-style SVD in embedding space* (Deerwester et al., 1990): stack the sentence embeddings for $S_j$, apply singular value decomposition, score sentences by contribution to the leading components, and concatenate the top $q$.

The resulting summary text $f_{\text{NLP}}^{(q)}(S_j)$ is then embedded in the same vector space as the documents to yield the new centroid $\boldsymbol{\mu}_j$.

## 3.2 K-LLMMEANS.

If instead we generate the summary with an LLM, we have k-LLMmeans. Formally, we replace equation 2 by

$$\boldsymbol{\mu}_j = \text{Embedding}(f_{\text{LLM}}(p_j)) \tag{4}$$

where $p_j = \text{Prompt}\left(I, \{d_{z_i} | z_i \sim [C_j]\}_{i=1}^{m_j}\right)$, and $m_j = \min(m, |C_j|)$. Here, $z_i \sim [C_j]$ denotes a sampled index of the embeddings assigned to cluster $C_j$ (without repetitions) and $m$ is a parameter that represents the maximum number of sampled indices used to compute the cluster centroid $\boldsymbol{\mu}_j$. In simple terms, we update a cluster's centroid by using the embedding of the response generated by an LLM when queried with a prompt containing a summarization instruction $I$ and a representative sample of documents from the cluster. Rather than providing all documents within the cluster as input, the LLM processes a representative sample as a context prompt. While incorporating the entire cluster is theoretically possible, it poses practical challenges due to prompt length limitations. Therefore, we propose selecting the sample cluster documents using a k-means++ sampling of the cluster embeddings. Our experiments demonstrate that this sampling process facilitates a more effective synthesis of the cluster's content, leading to improved summaries and, consequently, more refined centroid updates. The instruction $I$ varies depending on the clustering task, but standard summarization prompts are generally sufficient. Figure 1 illustrates how k-LLMmeans with a single summarization step enhances the standard k-means algorithm.

## 3.3 ADVANTAGES OF OUR APPROACHES

**Over k-means.** Periodic summarization steps act as a semantic prototype update that can redirect the search trajectory and reduce sensitivity to initialization. Instead of relying solely on Euclidean means, a textual summary captures contextual cues present in the underlying documents; re-embedding this summary yields centroids that better reflect cluster semantics. In practice, summary-based centroids produce more interpretable and often more semantically coherent partitions, even when k-means++ seeding is suboptimal. Apart from the summary steps, the procedure follows standard k-means: assignments and numeric updates are unchanged and the usual objective is preserved *between* summary iterations. Section 5.1 shows that our methods often outperform vanilla k-means across diverse settings.

**Over advanced LLM-based clustering methods.** Our approach offers three key advantages over more complex LLM-based clustering methods: (1) *Optimization landscape.* Grounded in the standard k-means objective, we inherit algorithmic convergence without relying on the fragile heuristics common in newer LLM-driven methods. A poor summary simply leads the procedure back toward a regular k-means local optimum, whereas competing approaches hinge on stable, format-specific LLM outputs. (2) *Scalability.* Unlike most state-of-the-art methods, whose LLM usage complexity grows with sample size (Feng et al., 2024; De Raedt et al., 2023) or require fine-tuning (Zhang et al., 2023). k-NLPmeans requires no LLM calls and k-LLMmeans only performs $k$ LLM calls per summarization step, with even few summarization steps yielding substantial performance gains. (See Section 5.1). (3) *Interpretability.* Replacing numeric centroids with textual summaries turns each prototype into a concise, human-readable synopsis; practitioners can track how cluster semantics evolve over time without post-hoc labeling. This transparency extends naturally to our mini-batch variant, enabling real-time monitoring in streaming scenarios (see Figure 2 and Section 5.1).

## 4 MINI-BATCH K-NLPMEANS AND K-LLMMEANS

Mini-batch k-means (Sculley, 2010) is an efficient strategy for large-scale text clustering that processes small, randomly sampled mini-batches instead of the full dataset. This approach substantially reduces memory usage and computational cost, making it well suited for continuously generated text

streams, such as those from social media, news, or customer feedback; where data must be clustered incrementally without full dataset access. Mini-batch k-means exhibits convergence properties comparable to standard k-means while offering superior scalability.

Although numerous streaming clustering methods that do *not* rely on LLMs have been studied (Silva et al., 2013; Aggarwal, 2018; Ribeiro et al., 2017; Aggarwal et al., 2003; Ackermann et al., 2012; Ordonez, 2003), only a few incorporate LLMs (Tarekegn et al., 2024; Nakshatri et al., 2023). Moreover, existing offline LLM-based clustering approaches face scalability issues, highlighting the need for scalable summary-based clustering in an online setting. To address this, we introduce *mini-batch k-NLPmeans and k-LLMmeans*, which directly extend mini-batch k-means by inserting summarization steps into the mini-batch update.

Algorithm 2 details how our approaches sequentially receive $b$ batches of documents $D_1, \ldots D_b$ where each batch contains a set of documents (these batches can either be random samples from a large corpus or represent sequential data). It processes each batch sequentially with k-NLPmeans/k-LLMmeans and updates centroids incrementally using a weighted rule like mini-batch k-means. Our algorithm preserves the desirable properties of mini-batch k-means, with low memory and none or low LLM usage. Section 6 shows that it also outperforms in simulations.

## 5 STATIC EXPERIMENTS

We use four benchmark datasets that span diverse domains and classification granularities: Bank77 (Casanueva et al., 2020), CLINC (Larson et al., 2019), GoEmo (Demszky et al., 2020) and MASSIVE (domain and intent) (FitzGerald et al., 2023). See Appendix A.1 for detailed descriptions of the datasets. We evaluate our algorithms on each of the four datasets using the known number of clusters and performing 120 centroid-update iterations. We calculate two variants of our algorithms differing in the number of summarization steps. The *single* variant uses a single summarization step ($l = 60$), while the *multiple* variant performs five summarization steps ($l = 20$). To demonstrate the robustness of our approach, we compute embeddings with models: DistilBERT(Sanh, 2019), e5-large(Wang et al., 2022), S-BERT(Reimers & Gurevych, 2019), and text-embedding-3-small(OpenAI, 2024). For k-NLPmeans we evaluate using *TextRank*, *Centroid* and *LSA* summarization methods mentioned in Section 3.1, with $q = 5$. For the LLM component of k-LLMmeans, we use GPT-3.5-turbo(OpenAI, 2023), GPT-4o(Hurst et al., 2024), Llama-3.3(Grattafiori et al., 2024), Claude-3.7(Anthropic, 2025) and DeepSeek-V3(Liu et al., 2024). For the instruction task $I$, we employ a simple summarization prompt depending on the task. For example, for Bank77 we use the prompt: *"The following is a cluster of online banking questions. Write a single question that represents the cluster concisely."* We explore the effect of prompt size by summarizing using all cluster documents as input, and a few-shot (*FS*) variant where each prompt includes only $m = 10$ randomly selected documents instead of the full cluster. For each setting we run five different seeds.

As traditional clustering baselines, we consider k-means, k-medoids (Kaufman & Rousseeuw, 2008), spectral clustering (Ng et al., 2001), agglomerative clustering (Johnson, 1967), and Gaussian Mixture Models (GMM) (Dempster et al., 1977). We use ground-truth number of clusters and ten different seeds. We also include BERTopic (Grootendorst, 2022) as a strong topic-modeling baseline. In addition, we report results obtained by applying our proposed methods to the document embeddings produced by BERTopic. For advanced LLM-based methods, we compare with results obtained by Feng et al. (2024) on ClusterLLM (Zhang et al., 2023), IDAS (De Raedt et al., 2023), and LLMEdgeRefine (Feng et al., 2024). All k-means based methods are initialized with k-means++.

We evaluate performance using clustering accuracy (ACC), which measures the best one-to-one alignment between predicted clusters and gold labels, and Normalized Mutual Information (NMI), which quantifies the mutual information between them normalized to $[0, 1]$ (See Appendix C for details on the metrics). We now provide a summary of the results.

### 5.1 RESULTS WITH STATIC DATA

**Comparing summarization variants and traditional clustering methods.** Table 1 reports mean accuracy (ACC) and normalized mutual information (NMI) for multiple k-NLPmeans and k-LLMmeans variants (LLM: GPT-4o), using text-embedding-3-small embeddings on four datasets.

Table 1: Average ACC and NMI for k-NLPmeans and k-LLMmeans variants using GPT-4o, compared against traditional baselines, BERTopic, and our k-NLPmeans *LSA-multiple* and k-LLMmeans *FS-multiple* variants applied to BERTopic embeddings, using text-embedding-3-small embeddings on benchmark datasets.

| Dataset/Method | BANK77 | | CLINC | | GoEmo | | Massive (D) | | Massive (I) | |
|---|---|---|---|---|---|---|---|---|---|---|
| | ACC | NMI | ACC | NMI | ACC | NMI | ACC | NMI | ACC | NMI |
| k-NLPmeans | | | | | | | | | | |
| *TextRank-single* | 66.4 | 83.6 | 78.7 | 92.4 | 21.5 | 20.9 | 60.6 | 68.9 | 53.0 | 72.5 |
| *TextRank-multiple* | 66.8 | 83.9 | 79.6 | 92.8 | 21.6 | 21.0 | 60.5 | 69.3 | 53.7 | 72.8 |
| *Centroid-single* | 67.2 | 84.0 | 78.5 | 92.2 | 22.1 | 21.2 | 60.9 | 68.3 | 53.5 | 73.0 |
| *Centroid-multiple* | 67.5 | 84.0 | 79.0 | 92.4 | 21.4 | 21.1 | 60.8 | 69.1 | 54.3 | 73.5 |
| *LSA-single* | 66.7 | 83.7 | 78.8 | 92.5 | 21.9 | 20.5 | 61.2 | 68.7 | 54.1 | 73.0 |
| *LSA-multiple* | 67.1 | 84.0 | 80.2 | 92.9 | 22.3 | 20.2 | 63.3 | 70.0 | 55.3 | 73.4 |
| k-LLMmeans | | | | | | | | | | |
| *single* | 67.1 | 83.6 | 78.1 | 92.5 | 24.0 | **22.3** | 61.1 | 69.6 | 54.0 | 73.0 |
| *multiple* | 66.7 | 83.6 | 79.1 | 92.8 | 24.1 | 22.1 | 60.6 | 69.5 | 55.8 | 73.5 |
| *FS-single* | 67.5 | 83.8 | 79.2 | 92.8 | 23.0 | 21.9 | 60.8 | 69.4 | 55.3 | 73.6 |
| *FS-multiple* | 67.9 | 84.1 | 80.2 | **93.1** | **24.2** | **22.3** | 63.2 | **70.6** | **56.7** | 73.9 |
| BERTopic | 71.2 | 83.0 | 79.0 | 91.6 | 18.2 | 16.6 | 36.1 | 61.6 | 54.8 | 70.6 |
| BERTopic+kNLPmeans | **76.0** | 87.3 | **82.1** | 93.0 | 17.4 | 17.2 | **65.0** | 70.2 | 56.4 | **74.3** |
| BERTopic+kLLMmeans | 75.7 | **87.4** | **82.1** | 93.0 | 17.7 | 17.2 | 63.9 | 69.8 | 56.4 | 74.2 |
| k-means | 66.2 | 83.0 | 77.3 | 92.0 | 20.7 | 20.5 | 59.4 | 67.9 | 52.9 | 72.4 |
| k-medoids | 41.7 | 69.5 | 49.3 | 77.7 | 15.7 | 15.9 | 37.6 | 38.8 | 35.9 | 52.4 |
| GMM | 67.7 | 83.0 | 78.9 | 92.7 | 21.5 | 20.6 | 56.2 | 68.6 | 53.9 | 73.2 |
| Agglomerative | 69.9 | 83.7 | 81.0 | 92.5 | 15.8 | 14.1 | 62.8 | 67.1 | 56.6 | 70.5 |
| Spectral | 68.2 | 83.3 | 76.3 | 90.9 | 17.6 | 15.2 | 61.5 | 67.0 | 56.6 | 71.4 |

Table 2: Average ACC, NMI, and $dist$ for k-means, k-NLPmeans *LSA-multiple* and k-LLMmeans *FS-multiple*, evaluated on three datasets using four different embedding models.

| Dataset | CLINC | | | GoEmo | | | Massive (D) | | |
|---|---|---|---|---|---|---|---|---|---|
| *embedding*/Method | ACC | NMI | $dist$ | ACC | NMI | $dist$ | ACC | NMI | $dist$ |
| *DistilBERT* | | | | | | | | | |
| k-means | 53.9 | 77.2 | **0.34** | 17.8 | 18.3 | 0.364 | 44.4 | 45.1 | 0.309 |
| k-NLPmeans | 52.9 | 77.3 | 0.353 | **18.5** | 18.2 | 0.365 | 44.7 | 45.3 | 0.311 |
| k-LLMmeans | **55.3** | **78.7** | 0.343 | 18.2 | **18.8** | **0.351** | **46.2** | **46.4** | **0.295** |
| *text-embedding-3-small* | | | | | | | | | |
| k-means | 77.3 | 92.0 | 0.2 | 20.7 | 20.5 | 0.287 | 59.4 | 67.9 | 0.246 |
| k-NLPmeans | 80.0 | 92.9 | **0.173** | 22.3 | 20.2 | 0.29 | **63.3** | 70.0 | **0.227** |
| k-LLMmeans | **80.2** | **93.1** | 0.179 | **24.2** | **22.3** | 0.278 | 63.2 | **70.6** | **0.227** |
| *e5-large* | | | | | | | | | |
| k-means | 73.8 | 90.8 | 0.131 | 22.8 | 22.8 | 0.176 | 58.4 | 63.7 | 0.138 |
| k-NLPmeans | 76.5 | 92.0 | **0.119** | 22.6 | 22.6 | 0.179 | 60.4 | 65.7 | 0.137 |
| k-LLMmeans | **77.2** | **92.5** | **0.119** | **24.2** | **24.3** | **0.168** | **62.3** | **65.9** | **0.133** |
| *S-BERT* | | | | | | | | | |
| k-means | 76.9 | 91.0 | 0.215 | 13.7 | 13.3 | 0.355 | 58.2 | 64.6 | 0.271 |
| k-NLPmeans | 79.0 | 91.9 | 0.201 | 14.1 | 12.8 | 0.363 | 58.5 | 64.6 | 0.272 |
| k-LLMmeans | **79.7** | **92.5** | **0.198** | **14.7** | **13.9** | **0.346** | **59.7** | **65.6** | **0.254** |

Across all datasets, our methods outperform traditional baselines on NMI and generally achieve higher ACC. Within k-NLPmeans, differences are modest, with LSA yielding the best overall scores among extractive summarizers. k-LLMmeans achieves the strongest results, attributable to higher-quality abstractive summaries; interestingly few-shot variants show better performance. Multiple summarization steps tend to offer additional gains over a single step. Overall, both k-NLPmeans and k-LLMmeans improve upon traditional algorithms, with k-LLMmeans delivering the best results, and remaining efficient, as few-shot summarization appears sufficient without needing extensive prompt length.

We also compare with BERTopic and apply our methods to its document embeddings. While BERTopic is strong on its own, adding our variants consistently improves its performance. Across all datasets, the best results come from either our stand-alone methods or their BERTopic-based versions.

Table 3: Number of LLM calls (prompts), average ACC, and average NMI for k-NLPmeans (*LSA-multiple*) and k-LLMmeans (*FS-multiple*) using various LLMs with e5-large embeddings, compared against BERTopic, our variants applied to BERTopic embeddings, and other state-of-the-art LLM-based clustering methods on three benchmark datasets.

| Dataset/Method | CLINC | | | GoEmo | | | Massive (D) | | |
|---|---|---|---|---|---|---|---|---|---|
| | prompts | ACC | NMI | prompts | ACC | NMI | prompts | ACC | NMI |
| k-NLPmeans | **0** | 76.5 | 92.0 | **0** | 22.6 | 22.6 | **0** | 60.4 | 65.7 |
| k-LLMmeans | | | | | | | | | |
|   *GPT-3.5-turbo* | 750 | 76.0 | 92.1 | 135 | 23.9 | 24.1 | 90 | **63.3** | 66.2 |
|   *GPT-4o* | 750 | 77.2 | 92.5 | 135 | 24.2 | 24.3 | 90 | 62.3 | 65.9 |
|   *Llama-3.3* | 750 | 77.2 | 92.3 | 135 | 23.8 | 23.1 | 90 | 62.0 | 66.3 |
|   *DeepSeek-V3* | 750 | 69.7 | 90.8 | 135 | 22.8 | 22.7 | 90 | 62.6 | 66.3 |
|   *Claude-3.7* | 750 | 76.9 | 92.5 | 135 | 24.2 | 23.7 | 90 | 61.8 | 66.0 |
| BERTopic | **0** | 77.8 | 90.8 | **0** | 20.3 | 20.8 | **0** | 38.5 | 61.7 |
| BERTopic+kNLPmeans | **0** | 81.5 | 93.0 | **0** | 23.2 | 22.6 | **0** | 60.8 | 66.1 |
| BERTopic+kLLMmeans | 750 | 81.4 | 93.0 | 135 | 24.0 | 23.0 | 90 | 58.6 | 65.6 |
| ClusterLLM | 1618 | 83.8 | 94.0 | 1618 | 26.8 | 23.9 | 1618 | 60.9 | **68.8** |
| IDAS | 4650 | 81.4 | 92.4 | 3011 | 30.6 | 25.6 | 2992 | 53.5 | 63.9 |
| LLMEdgeRefine | 1350 | **86.8** | **94.9** | 895 | **34.8** | **29.7** | 892 | 63.1 | 68.7 |

Table 4: Average ACC, and average NMI for four sequential mini-batch variants, k-means, mini-batch k-means, sequential mini-batch k-means on the yearly StackExchange data.

| Year/Method | 2020 (69147 posts) | | 2021 (54322 posts) | | 2022 (43521 posts) | | 2023 (38953 posts) | |
|---|---|---|---|---|---|---|---|---|
| | ACC | NMI | ACC | NMI | ACC | NMI | ACC | NMI |
| mini-batch k-NLPmeans | 68.0 | 79.5 | 67.9 | 78.5 | 69.0 | 78.9 | 71.6 | 78.8 |
| mini-batch k-LLMmeans | | | | | | | | |
|   *multiple* | 72.5 | 80.9 | **73.6** | **80.5** | **74.4** | **80.5** | **73.4** | **80.3** |
|   *FS + multiple* | **75.4** | **81.6** | 73.5 | 80.2 | 72.8 | 80.1 | 72.7 | 80.1 |
| k-means | 73.4 | 80.6 | 67.7 | 79.0 | 68.6 | 79.0 | 72.0 | 79.6 |
| mini-batch k-means | 67.0 | 78.2 | 67.7 | 77.4 | 67.5 | 77.6 | 67.2 | 77.0 |
| seq. mini-batch k-means | 67.0 | 76.6 | 66.7 | 75.2 | 65.6 | 75.6 | 65.8 | 74.8 |

**Comparing our approaches with k-means using different embeddings.** Table 2 compares k-NLPmeans (*LSA-multiple*) and k-LLMmeans (*FS-multiple*) with k-means with different embeddings on three benchmark datasets, reporting average ACC, average NMI, and average Euclidean distance between the learned and ground-truth centroids ($dist$). The $dist$ metric directly gauges how closely each algorithm recovers the true centroids, an especially meaningful criterion for centroid-based methods. Across all tested embeddings, our approaches achieve higher average ACC and NMI than k-means producing smaller average $dist$ values. Interestingly k-LLMmeans stands as the best, demonstrating that its LLM-guided centroid updates converge to solutions that are both more accurate and more faithfully aligned with the underlying cluster structure.

**Comparing our approaches with different LLMs and state-of-the-art LLM-based clustering methods.** Table 3 reports the number of LLM calls (prompts) and mean ACC/NMI for k-NLPmeans (*LSA-multiple*) and k-LLMmeans (*FS-multiple*) on three benchmarks with e5-large embeddings. For k-LLMmeans we vary the LLM generator, including GPT-3.5, and observe stable performance across LLMs, indicating robustness to the specific model. By design, k-NLPmeans uses zero LLM calls. For context, we also include results from Feng et al. (2024): ClusterLLM (fine-tuned) and IDAS use GPT-3.5 with the same e5-large embeddings, while LLMEdgeRefine uses the stronger Instructor embeddings (Jin et al., 2023). Our GPT-3.5 configuration of k-LLMmeans achieves comparable, sometimes slightly lower ACC/NMI, but with much fewer LLM calls (independent of dataset size) and no fine-tuning. k-NLPmeans trails k-LLMmeans slightly but remains competitive without any LLM usage. We also evaluate BERTopic and apply our variants to its embeddings. In this setting, our methods achieve results extremely close to the best LLM-based models while requiring none or only a small number of LLM calls. Overall, our framework offers a favorable quality–cost trade-off: k-NLPmeans for zero-inference-cost deployments, and k-LLMmeans for higher accuracy under a small, fixed summarization budget.

**Additional results and experiments.** Results with standard deviations are reported in Tables 5, 6, and 7 of Appendix B.1. We also assess sensitivity to the selection of the number of clusters $k$ in Appendix B.2.1. Table 9 shows robustness when the number of clusters mildly differs from the ground truth clusters, consistently better than vanilla k-means. We also evaluate sensitivity to the instruction prompt $I$ in k-LLMmeans and to parameter $q$ in k-NLPmeans in Appendix B.2.2. As shown in Table 10, performance remains generally stable across prompt choices and $q$ values. Table 11 in Appendix B.2.3 also compares several sampling strategies for the few-shot (FS) version of k-LLMmeans, showing that our k-means++ strategy has consistent performance.

**Summary of static experimental results** Across our experiments, our methods (i) improve over classical and topic-modeling baselines (Table 1); (ii) outperforms k-means across embeddings (Table 2); and (iii) approaches strong LLM-based clustering methods at much lower LLM cost (Table 3).

## 6 SEQUENTIAL (MINI-BATCH) EXPERIMENTS

We use yearly posts from 2020 to 2023 from 35 StackExchange sites (StackExchange, 2024). Each post is accompanied by the site label (domain) and timestamp, making this dataset well-suited for evaluating online or sequential clustering methods. We release this dataset alongside our submission (see Appendix A.2 for details). For each yearly subset, we split the data into $b = \lceil \frac{n}{10000} \rceil$ equal-sized batches $D_1, \ldots, D_b$ in chronological order, where $n$ is the number of documents for that year. We run the mini-batch k-LLMmeans algorithm in its four variants described in Section 5 (for practical reasons we set $m = 50$ for the full cluster variant). We compare with three baselines: mini-batch k-means with standard random sampling, sequential mini-batch k-means with $b$ chronological batches, and standard k-means on the full dataset. We use ground-truth clusters, text-embedding-3-small for embeddings, GPT-4o for the summarization step, and five different seeds.

### 6.1 RESULTS WITH STREAMING DATA

Table 4 reports average ACC and NMI on the yearly StackExchange corpus, comparing mini-batch k-NLPmeans (LSA-multiple) and mini-batch k-LLMmeans variants against standard k-means, mini-batch k-means, and sequential mini-batch k-means. Mini-batch k-LLMmeans variants achieve the highest ACC and NMI, consistently outperforming baselines, and even surpassing full-dataset k-means. Despite operating in a streaming regime, our approach clusters the entire corpus of 205,943 posts using no more than 3,850 LLM calls, demonstrating a favorable accuracy-per-LLM-token trade-off. The few-shot summarization design keeps prompts short, sidestepping context-window limits while preserving cluster quality. Mini-batch k-NLPmeans also improves over all mini-batch baselines, though it trails the k-LLMmeans variants, likely because its extractive summaries are more sensitive to the noisiness and heterogeneity of community posts than LLM-generated centroids. Overall, these results highlight that our interpretable, mini-batch formulation scales to long sequential streams with strong accuracy and limited LLM usage.

Results with standard deviations are reported in Table 8 (Appendix B.1).

## 7 CASE STUDY

To demonstrate the interpretability of our method in capturing the evolution of clusters within sequential data, we present a case study using posts from the AI site in the 2021 Stack Exchange dataset (StackExchange, 2024). We apply our mini-batch k-LLMmeans algorithm with three equal-length batches and a total of ten clusters. We use the instruction *"The following is a cluster of questions from the AI community. Write a single question that represents the cluster"*.

**Interpretation.** Our interpretable clustering enables temporal analysis of topic change, since each centroid is a human-readable summary rather than a latent vector. Figure 2 shows this for three major clusters: *Image Model Optimization*, *AI Evolution and Challenges*, and *Advanced NLP Techniques*. Across 2021, these clusters drift from fundamentals toward integration-at-scale, mirroring how tools, models, and deployment pressures matured. *Image Model Optimization* begins with robustness anxieties (adversaries, occlusion, scale) around mainstream detectors/segmenters, pivots mid-year to "how do I implement this well?" (transfer learning, augmentation, preprocessing), and

| Cluster description | First batch | Second batch | Third batch |
|---|---|---|---|
| **Image Model Optimization** | How can we improve the robustness and accuracy of image classification and object detection models, such as CNNs, YOLO, and Mask R-CNN, against challenges like adversarial attacks, occlusion, and varying input scales, while also effectively utilizing additional data types and addressing issues like overfitting, segmentation, and bounding box precision? | How can I effectively implement and optimize various deep learning techniques, such as transfer learning, object detection, and image preprocessing, for tasks like facial expression recognition, semantic segmentation, and anomaly detection, while addressing challenges like input size inconsistency, data augmentation, and model uncertainty? | How can I effectively utilize AI techniques, such as convolutional neural networks (CNNs) and object detection models, for tasks involving image data, including detecting specific objects or features, handling imbalanced datasets, and improving model performance through techniques like transfer learning, data augmentation, and specialized loss functions? |
| **AI evolution and challenges** | How has artificial intelligence research evolved in various domains, such as board games, semantic networks, and computational theories, and what are the practical implications and challenges of implementing AI in areas like gaming, formal logic, and general intelligence? | How do current AI models and algorithms, particularly in the context of AGI and generative models like GANs, incorporate or benefit from the concept of flaws or limitations, and what philosophical or practical implications do these imperfections have for achieving human-level intelligence or solving complex problems? | How can different AI approaches, such as symbolic AI, neural networks, and hybrid systems, be effectively utilized or combined to achieve AGI, considering factors like computational efficiency, and the integration of various AI techniques, including knowledge engineering, neuro-symbolic methods, and reinforcement learning, while addressing challenges related to safety, scalability, and adaptability in diverse applications? |
| **Advanced NLP Techniques** | How can I effectively utilize NLP techniques and models, such as Word2Vec, BERT, and Transformers, for various tasks like word embedding, sentiment analysis, text classification, and handling grammatical errors, while understanding the differences between these models and the role of preprocessing in improving their performance? | How can we effectively evaluate and understand the contextual and semantic capabilities of models like BERT and GPT-3, considering their use of embeddings, self-attention mechanisms, and transfer learning, while also exploring alternative evaluation metrics beyond traditional ones like BLEU for tasks such as text generation and translation? | How can I effectively utilize pre-trained language models and NLP techniques to handle tasks such as text translation, entity recognition, and text classification, while addressing challenges like sequence length limitations, domain-specific vocabulary, and the need for accurate alignment between text and audio in multilingual contexts? |

Figure 2: Sequential evolution of the LLM-generated centroids for three primary clusters during the three batches of the sequential mini-batch k-LLMmeans process applied to 2021 posts from the AI Stack Exchange site (StackExchange, 2024). Main aspects are manually highlighted.

ends with production hardening (class imbalance, specialized losses, edge efficiency) as strong pre-trained models and libraries made setup routine and pushed bottlenecks to data and deployment (Bochkovskiy et al., 2020; Kolesnikov et al., 2021). *AI Evolution and Challenges* moves from historical "what is AI becoming?" to curiosity about large generative/multimodal systems, probably sparked by public releases like CLIP/DALL·E and their failure modes, and finishes with system-level synthesis (neuro-symbolic integration, RL control, compute efficiency, safety) as enterprise adoption and policy attention grow (Radford et al., 2021; Ramesh et al., 2021; Nayak, 2021; OpenAI, 2021; European Commission, 2021). *Advanced NLP Techniques* tracks the transformer boom: early questions focus on which models and preprocessing to use; mid-year attention shifts to measuring semantics beyond BLEU (e.g., BERTScore, BLEURT) as long-context and domain-shift issues surface; late-year concerns are product-driven, sequence limits, domain vocab, multilingual/cross-modal alignment; reflecting rapid uptake of pretrained encoders/decoders in industry (Brown et al., 2020; Zhang et al., 2020b; Sellam et al., 2020; Xue et al., 2021). Practically, these trajectories can power better post categorization, searchability, answer routing, and trend detection. More broadly, they showcase how our mini-batch k-LLMmeans exposes readable centroids enabling end-to-end interpretability for sequential text streams, letting practitioners track topic evolution, attribute causes, and make transparent, timely decisions in dynamic corpora.

## 8 RELATED WORK

**Traditional clustering.** Hierarchical methods (Johnson, 1967; Blashfield & Aldenderfer, 1978) build tree-structured representations of nested document relationships. Density-based approaches like DBSCAN (Ester et al., 1996) and graph-based methods detect clusters of arbitrary shapes, while spectral clustering (Ng et al., 2001) leverages eigen-decomposition to uncover complex structures. Model-based techniques; including Gaussian mixture models (Dempster et al., 1977) and recent neural network frameworks (Zhou et al., 2019; Huang et al., 2014; Yang et al., 2016; Zhang et al., 2021; Xie et al., 2016); provide probabilistic clustering formulations. Additionally, topic modeling methods, from probabilistic latent semantic analysis (Hofmann, 2001) to latent Dirichlet allocation (Blei et al., 2003), capture word co-occurrence patterns and latent topics. BERTopic (Grootendorst, 2022) offers an embedding–reduction–density pipeline for interpretable topics, while DeepCluster (Caron et al., 2018) uses clustering to iteratively train encoders in a self-supervised manner (orthogonal to our fixed-embedding setting). Jia & Diaz-Rodriguez (2026) also incorporate modern embeddings into text segmentation within classical change point detection methods.

**LLM-based clustering.** Viswanathan et al. (2024) employ LLMs to augment document representations, generate pseudo pairwise constraints, and post-correct low-confidence assignments for query-efficient, few-shot semi-supervised clustering. Zhang et al. (2023) propose ClusterLLM, which uses instruction-tuned LLMs via interactive triplet and pairwise feedback to cost-effectively

refine clustering granularity. Complementary approaches (Tipirneni et al., 2024; Petukhova et al., 2025) show that context-derived representations capture subtle semantic nuances beyond traditional embeddings. Wang et al. (2023) introduce a goal-driven, explainable method that employs natural language descriptions to clarify cluster boundaries, while (De Raedt et al., 2023) present IDAS for intent discovery using abstractive summarization. Feng et al. (2024) propose LLMEdgeRefine an iterative mechanism that forms super-points to mitigate outliers and reassign ambiguous edge points, resulting in clusters with higher coherence and robustness.

## 9  DISCUSSION

We presented k-NLPmeans and k-LLMmeans, which replace the numeric centroid in k-means with textual summaries, yielding interpretable prototypes while preserving scalability. The LLM-free k-NLPmeans achieves competitive accuracy with minimal additional complexity; k-LLMmeans augments clusters with LLM summaries under a fixed, iteration-bounded LLM budget, offering a favorable accuracy–efficiency trade-off without fine-tuning. A mini-batch extension enables streaming operation. Overall, *summary-as-centroid* is a simple yet powerful and novel modification that unifies interpretability and efficiency and broadens the applicability of k-means to modern text streams. **Beyond text**, *summary-as-centroid* could be generalized by replacing numeric centroids with re-embedded, domain-specific prototypes (e.g., image collages, synthetic utterances, DNA consensus motifs, representative subgraphs), preserving interpretability without altering the clustering.

**Limitations.** Our k-LLMmeans relies on LLM-generated summaries; thus, biases or errors in the underlying model can propagate to cluster prototypes and downstream analyses. Although simple instruction prompts are effective (Appendix B.2), prompt design can still influence outcomes. The few-shot variant assumes that a small set of representative samples captures cluster structure, practical in many settings but potentially limiting for heterogeneous clusters. Regarding potential pretraining exposure, note that the LLM in our pipeline is used solely to summarize texts already assigned to each cluster; it does not inject external labels or unseen content, so any exposure does not confer an unfair advantage. As indicated by suboptimal results on the StackExchange dataset (Table 4), k-NLPmeans may benefit from applying text-cleaning preprocessing prior to use. Finally, we illustrate qualitative failure cases in Appendix E.

**Cost.** Our k-NLPmeans is LLM-free; runtime is dominated by standard k-means and a lightweight summarization step. In contrast, k-LLMmeans incurs one LLM call per cluster per summarization step. These calls are highly parallelizable, so wall-clock time is roughly the latency of a single LLM call multiplied by the number of summarization steps. In practice, running any static dataset with five summarization rounds using GPT-4o and text-embedding-3-small cost $< \$1$ and completed in $\approx 1$ minute on a single laptop without parallelization; for the Stack Overflow dataset, the end-to-end run cost $2.5 and took $\approx 8$ minutes under the same conditions. By comparison, competing LLM-heavy approaches would require $18–$25 and $> 40$ minutes on identical hardware and API settings. Thus, k-LLMmeans remains interpretable and cost-efficient among LLM-assisted methods, while k-NLPmeans provides a strong, interpretable, zero-LLM alternative when budget or latency is constrained.

## REPRODUCIBILITY STATEMENT

We release data and code to reproduce all results in this paper at `https://github.com/jairoadiazr/summaryCentroids` (see the README for instructions).

## ACKNOWLEDGMENTS

This work was supported by the Natural Sciences and Engineering Research Council of Canada (NSERC) under grant DGECR-2022-04531. The author thanks Mumin Jia, the anonymous reviewers, and the session chair for their valuable feedback and insightful suggestions, which greatly improved the quality of this work.

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

## A DATASETS

### A.1 NON-STREAMING DATASETS

We evaluate our clustering approach on four benchmark datasets. **Bank77** (Casanueva et al., 2020): Consists of 3,080 customer queries related to banking services, categorized into 77 distinct intents. **CLINC** (Larson et al., 2019): A diverse set of 4,500 queries spanning 150 intent classes across multiple domains, designed for open-domain intent classification. **GoEmo** (Demszky et al., 2020): Contains 2,984 social media posts annotated with 27 fine-grained emotion categories. We removed the neutral expressions to address data imbalance and retained only entries with a single, unique emotion. **MASSIVE** (FitzGerald et al., 2023): Comprises 2,974 English-language virtual assistant utterances grouped into 18 domains and 59 intent categories.

These datasets provide a robust evaluation setting for text clustering across different domains and classification granularities.

## A.2 New compiled dataset for testing text-Streaming Clustering Algorithms

We extract and unify a challenging data stream comprising unique archive posts collected from 84 Stack Exchange sites (StackExchange, 2024). Each post is accompanied by the site label (domain) and timestamp, making this dataset well-suited for evaluating online or sequential clustering methods. Our raw dataset spans 84 domains, each containing at least 20 posts per year from 2018 to 2023 (with post lengths ranging from 20 to 1000 characters), totaling 499,359 posts. For our experiments, we focus on posts from 2020 to 2023 and further filter out labels that do not exceed 500 posts in 2023. The resulting subset comprises 35 distinct groups and 69,147 posts. Both the raw and clean data are provided with this paper. Stack Exchange content is licensed under the Creative Commons Attribution-ShareAlike 4.0 International (CC BY-SA 4.0) license.

## B Supplementary results

### B.1 Main results with standard deviations

In Tables 5, 6, 7 and 4 we report the same results as in Tables 1, 2, 3 and 8 including standard deviations.

### B.2 Additional experiments

#### B.2.1 Sensitivity to number of clusters $k$

We also examined the sensitivity to the parameter $k$ (the number of clusters) by assessing the effect when the number of clusters $k$ is set below the ground truth ($k-20\%$, $k-10\%$), at the ground truth ($k$), and above it ($k+10\%$, $k+20\%$). The corresponding results are reported in Table 9. As shown there, both k-NLPmeans and k-LLMmeans consistently outperform kmeans regardless of the estimation error in $k$, and the overall impact on ACC and NMI remains minor.

#### B.2.2 Sensitivity to prompt $I$ and parameter $q$

We evaluated the sensitivity of k-LLMmeans to the instruction prompt $I$ on BANK77 by testing five prompt variants using k-LLMmeans (FS-single) with text-embedding-3-small and GPT-4o. We also examined the effect of the parameter $q$ on the same dataset and embeddings. As shown in Table 10, performance remains remarkably stable across prompt choices and $q$ settings, indicating that the methods are insensitive to these hyperparameters.

#### B.2.3 Sensitivity to number sampling strategy for the few-shot (FS) variant of k-LLMmeans

We also examine the effect of alternative sampling procedures for the few shot variant. We propose in the manuscript k-means++ but here we also explore uniform random selection (random), choosing closest to the cluster centroid (centroid) and choosing the farthest ones (edge). The corresponding results are reported in Table 11. As shown there, our proposed k-means++ seems to have a consistent performance.

## C Evaluation metrics

For evaluation, we report clustering accuracy (ACC) and normalized mutual information (NMI). NMI is computed with `normalized_mutual_info_score` from `sklearn.metrics`, which implements the standard mutual-information–based normalization. Let $Y$ denote the ground-truth labels and $\hat{Y}$ the predicted cluster labels. We first form the contingency matrix $W \in \mathbb{N}^{D \times D}$ with entries $W_{ij} = |\{n : \hat{y}_n = i,\ y_n = j\}|$, where $D$ is the maximum label index plus one and $N$ is the number of data points. The empirical mutual information between $Y$ and $\hat{Y}$ is

$$\mathrm{MI}(Y, \hat{Y}) \;=\; \sum_{i,j} \frac{W_{ij}}{N} \log \frac{N\, W_{ij}}{\left(\sum_{j'} W_{ij'}\right)\left(\sum_{i'} W_{i'j}\right)},$$

Table 5: Average ACC and NMI for k-NLPmeans and k-LLMmeans variants using GPT-4o, compared against traditional baselines, BERTopic, and our k-NLPmeans *LSA-multiple* and k-LLMmeans *FS-multiple* variants applied to BERTopic embeddings, using text-embedding-3-small embeddings on benchmark datasets. Standard deviations of ACC and NMI in parenthesis

| Dataset/Method | BANK77 | | CLINC | | GoEmo | | Massive (D) | | Massive (I) | |
|---|---|---|---|---|---|---|---|---|---|---|
| | ACC | NMI | ACC | NMI | ACC | NMI | ACC | NMI | ACC | NMI |
| k-NLPmeans | | | | | | | | | | |
| *TextRank-single* | 66.4 | 83.6 | 78.7 | 92.4 | 21.5 | 20.9 | 60.6 | 68.9 | 53.0 | 72.5 |
| | (0.94) | (0.47) | (1.56) | (0.41) | (0.63) | (0.29) | (3.47) | (1.27) | (1.8) | (0.71) |
| *TextRank-multiple* | 66.8 | 83.9 | 79.6 | 92.8 | 21.6 | 21.0 | 60.5 | 69.3 | 53.7 | 72.8 |
| | (0.94) | (0.45) | (1.51) | (0.45) | (1.29) | (0.65) | (3.66) | (1.22) | (1.65) | (0.58) |
| *Centroid-single* | 67.2 | 84.0 | 78.5 | 92.2 | 22.1 | 21.2 | 60.9 | 68.3 | 53.5 | 73.0 |
| | (0.81) | (0.39) | (1.68) | (0.37) | (0.6) | (0.41) | (3.02) | (0.91) | (2.07) | (1.13) |
| *Centroid-multiple* | 67.5 | 84.0 | 79.0 | 92.4 | 21.4 | 21.1 | 60.8 | 69.1 | 54.3 | 73.5 |
| | (0.57) | (0.3) | (1.95) | (0.46) | (0.87) | (0.12) | (2.72) | (1.09) | (1.72) | (0.84) |
| *LSA-single* | 66.7 | 83.7 | 78.8 | 92.5 | 21.9 | 20.5 | 61.2 | 68.7 | 54.1 | 73.0 |
| | (0.99) | (0.47) | (2.07) | (0.63) | (1.11) | (0.87) | (2.66) | (1.03) | (1.86) | (0.91) |
| *LSA-multiple* | 67.1 | 84.0 | 80.2 | 92.9 | 22.3 | 20.2 | 63.3 | 70.0 | 55.3 | 73.4 |
| | (0.98) | (0.42) | (0.98) | (0.41) | (0.85) | (0.56) | (3.06) | (1.02) | (1.32) | (0.72) |
| k-LLMmeans | | | | | | | | | | |
| *single* | 67.1 | 83.6 | 78.1 | 92.5 | 24.0 | **22.3** | 61.1 | 69.6 | 54.0 | 73.0 |
| | (1.22) | (0.46) | (1.67) | (0.47) | (1.35) | (0.57) | (3.39) | (1.6) | (1.51) | (0.58) |
| *multiple* | 66.7 | 83.6 | 79.1 | 92.8 | 24.1 | 22.1 | 60.6 | 69.5 | 55.8 | 73.5 |
| | (0.78) | (0.2) | (1.68) | (0.48) | (1.33) | (0.65) | (2.82) | (1.66) | (1.4) | (0.76) |
| *FS-single* | 67.5 | 83.8 | 79.2 | 92.8 | 23.0 | 21.9 | 60.8 | 69.4 | 55.3 | 73.6 |
| | (1.07) | (0.36) | (1.73) | (0.43) | (0.94) | (0.64) | (3.21) | (1.09) | (1.76) | (0.77) |
| *FS-multiple* | 67.9 | 84.1 | 80.2 | **93.1** | **24.2** | **22.3** | 63.2 | **70.6** | **56.7** | 73.9 |
| | (1.55) | (0.41) | (1.52) | (0.32) | (1.16) | (0.67) | (2.84) | (1.38) | (1.32) | (0.7) |
| BERTopic | 71.2 | 83.0 | 79.0 | 91.6 | 18.2 | 16.6 | 36.1 | 61.6 | 54.8 | 70.6 |
| | (0.0) | (0.0) | (0.0) | (0.0) | (0.0) | (0.0) | (0.0) | (0.0) | (0.0) | (0.0) |
| BERTopic+kNLPmeans | **76.0** | 87.3 | **82.1** | 93.0 | 17.4 | 17.2 | **65.0** | 70.2 | 56.4 | **74.3** |
| | (0.76) | (0.11) | (0.84) | (0.09) | (0.7) | (0.3) | (2.39) | (0.66) | (1.86) | (0.27) |
| BERTopic+kLLMmeans | 75.7 | **87.4** | **82.1** | 93.0 | 17.7 | 17.2 | 63.9 | 69.8 | 56.4 | 74.2 |
| | (1.26) | (0.17) | (0.83) | (0.2) | (0.43) | (0.3) | (1.35) | (0.63) | (1.19) | (0.12) |
| k-means | 66.2 | 83.0 | 77.3 | 92.0 | 20.7 | 20.5 | 59.4 | 67.9 | 52.9 | 72.4 |
| | (1.82) | (0.56) | (1.83) | (0.35) | (1.0) | (0.67) | (4.03) | (1.67) | (1.07) | (0.54) |
| k-medoids | 41.7 | 69.5 | 49.3 | 77.7 | 15.7 | 15.9 | 37.6 | 38.8 | 35.9 | 52.4 |
| | (0.0) | (0.0) | (0.0) | (0.0) | (0.0) | (0.0) | (0.0) | (0.0) | (0.0) | (0.0) |
| GMM | 67.7 | 83.0 | 78.9 | 92.7 | 21.5 | 20.6 | 56.2 | 68.6 | 53.9 | 73.2 |
| | (1.78) | (0.54) | (1.18) | (0.22) | (0.82) | (0.47) | (2.88) | (1.01) | (1.91) | (0.80) |
| Agglomerative | **69.9** | 83.7 | **81.0** | 92.5 | 15.8 | 14.1 | 62.8 | 67.1 | 56.6 | 70.5 |
| | (0.0) | (0.0) | (0.0) | (0.0) | (0.0) | (0.0) | (0.0) | (0.0) | (0.0) | (0.0) |
| Spectral | 68.2 | 83.3 | 76.3 | 90.9 | 17.6 | 15.2 | 61.5 | 67.0 | 56.6 | 71.4 |
| | (0.56) | (0.38) | (0.401) | (0.14) | (0.32) | (0.32) | (0.02) | (0.02) | (0.84) | (0.23) |

and the entropies are

$$H(Y) = -\sum_j \frac{\sum_i W_{ij}}{N} \log \frac{\sum_i W_{ij}}{N}, \qquad H(\hat{Y}) = -\sum_i \frac{\sum_j W_{ij}}{N} \log \frac{\sum_j W_{ij}}{N}.$$

Following the default in `sklearn`, NMI is then defined as

$$\mathrm{NMI}(Y, \hat{Y}) = \frac{\mathrm{MI}(Y, \hat{Y})}{\frac{1}{2}\big(H(Y) + H(\hat{Y})\big)}.$$

ACC is defined as the fraction of correctly assigned points after the best one-to-one relabeling of clusters. Concretely, given the same contingency matrix $W$, we solve a maximum-weight bipartite matching using the Hungarian algorithm on $W$ and compute

$$\mathrm{ACC} = \frac{1}{N} \sum_{(i,j) \in \mathcal{M}} W_{ij},$$

where $\mathcal{M}$ is the set of matched label pairs returned by the Hungarian algorithm.

Table 6: Average ACC, NMI, and $dist$ for k-means, k-NLPmeans *LSA-multiple* and k-LLMmeans *FS-multiple*, evaluated on three datasets using four different embedding models. Standard deviations of ACC, NMI and $dist$ in parenthesis.

| Dataset | CLINC | | | GoEmo | | | Massive (D) | | |
|---|---|---|---|---|---|---|---|---|---|
| *embedding*/Method | ACC | NMI | $dist$ | ACC | NMI | $dist$ | ACC | NMI | $dist$ |
| *DistilBERT* | | | | | | | | | |
| k-means | 53.9 | 77.2 | **0.34** | 17.8 | 18.3 | 0.364 | 44.4 | 45.1 | 0.309 |
| | (0.96) | (0.49) | (0.007) | (0.86) | (0.67) | (0.005) | (1.53) | (0.99) | (0.021) |
| k-NLPmeans | 52.9 | 77.3 | 0.353 | **18.5** | 18.2 | 0.365 | 44.7 | 45.3 | 0.311 |
| | (0.25) | (0.22) | (0.003) | (0.58) | (0.38) | (0.004) | (2.44) | (1.22) | (0.02) |
| k-LLMmeans | **55.3** | **78.7** | 0.343 | 18.2 | **18.8** | **0.351** | **46.2** | **46.4** | **0.295** |
| | (1.1) | (0.53) | (0.009) | (0.52) | (0.73) | (0.004) | (1.35) | (1.12) | (0.015) |
| *text-embedding-3-small* | | | | | | | | | |
| k-means | 77.3 | 92.0 | 0.2 | 20.7 | 20.5 | 0.287 | 59.4 | 67.9 | 0.246 |
| | (1.83) | (0.35) | (0.014) | (1.0) | (0.67) | (0.005) | (4.03) | (1.67) | (0.015) |
| k-NLPmeans | **80.2** | 92.9 | **0.173** | 22.3 | 20.2 | 0.29 | **63.3** | 70.0 | **0.227** |
| | (0.98) | (0.41) | (0.006) | (0.85) | (0.56) | (0.007) | (3.06) | (1.02) | (0.018) |
| k-LLMmeans | **80.2** | **93.1** | 0.179 | **24.2** | **22.3** | **0.278** | 63.2 | **70.6** | **0.227** |
| | (1.52) | (0.32) | (0.015) | (1.16) | (0.67) | (0.011) | (2.84) | (1.38) | (0.022) |
| *e5-large* | | | | | | | | | |
| k-means | 73.8 | 90.8 | 0.131 | 22.8 | 22.8 | 0.176 | 58.4 | 63.7 | 0.138 |
| | (2.34) | (0.74) | (0.01) | (0.92) | (0.73) | (0.003) | (3.89) | (1.75) | (0.02) |
| k-NLPmeans | 76.5 | 92.0 | **0.119** | 22.6 | 22.6 | 0.179 | 60.4 | 65.7 | 0.137 |
| | (1.76) | (0.58) | (0.009) | (0.7) | (0.32) | (0.003) | (3.2) | (1.79) | (0.008) |
| k-LLMmeans | **77.2** | **92.5** | **0.119** | **24.2** | **24.3** | **0.168** | **62.3** | **65.9** | **0.133** |
| | (1.6) | (0.4) | (0.007) | (0.76) | (0.7) | (0.003) | (1.36) | (0.78) | (0.007) |
| *S-BERT* | | | | | | | | | |
| k-means | 76.9 | 91.0 | 0.215 | 13.7 | 13.3 | 0.355 | 58.2 | 64.6 | 0.271 |
| | (1.63) | (0.5) | (0.013) | (0.62) | (0.57) | (0.005) | (2.03) | (1.09) | (0.023) |
| k-NLPmeans | 79.0 | 91.9 | 0.201 | 14.1 | 12.8 | 0.363 | 58.5 | 64.6 | 0.272 |
| | (1.65) | (0.31) | (0.017) | (0.3) | (0.29) | (0.003) | (2.24) | (1.29) | (0.022) |
| k-LLMmeans | **79.7** | **92.5** | **0.198** | **14.7** | **13.9** | **0.346** | **59.7** | **65.6** | **0.254** |
| | (0.73) | (0.25) | (0.008) | (0.55) | (0.5) | (0.004) | (2.54) | (0.79) | (0.019) |

# D  ALGORITHMS

This section includes Algorithm 1 (k-NLPmeans / k-LLMmeans) and Algorithm 2 (mini batch version).

# E  QUALITATIVE ANALYSIS AND FAILURE MODES

In the main paper we focused on successful examples to illustrate how summary-as-centroid can make clusters more interpretable. For completeness, we provide qualitative *failure* cases that highlight typical limitations of our approach. The examples below are representative, not exhaustive, and are lightly anonymized / paraphrased for clarity.

## E.1  PROMPT-ECHO AND META SUMMARIES

The first failure mode we mention (for k-LLMmeans) is "prompt echoing", where the LLM partially repeats the instruction instead of summarizing the actual content.

**Example 1 (Prompt-echo / meta summary).**

---

[1]Here k-NLPMmeans/k-LLMmeans is initialized with the final centroids of the previous batch

Table 7: Number of LLM calls (prompts), average ACC, and average NMI for k-NLPmeans (*LSA-multiple*) and k-LLMmeans (*FS-multiple*) using various LLMs with e5-large embeddings, compared against BERTopic, our variants applied to BERTopic embeddings, and other state-of-the-art LLM-based clustering methods on three benchmark datasets. Standard deviations of ACC and NMI in parenthesis (not available for baseline LLM-based methods).

| Dataset/Method | CLINC | | | GoEmo | | | Massive (D) | | |
|---|---|---|---|---|---|---|---|---|---|
| | prompts | ACC | NMI | prompts | ACC | NMI | prompts | ACC | NMI |
| k-NLPmeans | **0** | 76.5 | 92.0 | **0** | 22.6 | 22.6 | **0** | 60.4 | 65.7 |
| | | (0.02) | (0.01) | | (0.01) | (0.0) | | (0.03) | (0.02) |
| k-LLMmeans | | | | | | | | | |
| *GPT-3.5* | 750 | 76.0 | 92.1 | 135 | 23.9 | 24.1 | 90 | 63.3 | 66.2 |
| | | (1.16) | (0.28) | | (0.65) | (0.39) | | (1.47) | (0.62) |
| *GPT-4o* | 750 | 77.2 | 92.5 | 135 | 24.2 | 24.3 | 90 | 62.3 | 65.9 |
| | | (1.6) | (0.4) | | (0.76) | (0.7) | | (1.36) | (0.78) |
| *Llama-3.3* | 750 | 77.2 | 92.3 | 135 | 23.8 | 23.1 | 90 | 62.0 | 66.3 |
| | | (1.6) | (0.44) | | (0.66) | (0.62) | | (2.97) | (1.56) |
| *DeepSeek-V3* | 750 | 69.7 | 90.8 | 135 | 22.8 | 22.7 | 90 | 62.6 | 66.3 |
| | | (1.02) | (0.29) | | (1.68) | (0.88) | | (2.24) | (0.71) |
| *Claude-3.7* | 750 | 76.9 | 92.5 | 135 | 24.2 | 23.7 | 90 | 61.8 | 66.0 |
| | | (1.83) | (0.52) | | (1.0) | (0.65) | | (4.83) | (1.99) |
| BERTopic | **0** | 77.8 | 90.8 | **0** | 20.3 | 20.8 | **0** | 38.5 | 61.7 |
| | | (0.0) | (0.0) | | (0.0) | (0.0) | | (0.0) | (0.0) |
| BERTopic+kNLPmeans | **0** | 81.5 | 93.0 | **0** | 23.2 | 22.6 | **0** | 60.8 | 66.1 |
| | | (0.78) | (0.18) | | (0.62) | (0.13) | | (1.42) | (0.58) |
| BERTopic+kLLMmeans | 750 | 81.4 | 93.0 | 135 | 24.0 | 23.0 | 90 | 58.6 | 65.6 |
| | | (0.55) | (0.11) | | (0.36) | (0.21) | | (2.96) | (0.98) |
| ClusterLLM | 1618 | 83.8 | 94.0 | 1618 | 26.8 | 23.9 | 1618 | 60.9 | **68.8** |
| IDAS | 4650 | 81.4 | 92.4 | 3011 | 30.6 | 25.6 | 2992 | 53.5 | 63.9 |
| LLMEdgeRefine | 1350 | **86.8** | **94.9** | 895 | **34.8** | **29.7** | 892 | 63.05 | 68.67 |

Table 8: Average ACC, and average NMI for four sequential mini-batch variants, k-means, mini-batch k-means, sequential mini-batch k-means on the yearly StackExchange data. Standard deviations of ACC and NMI in parenthesis.

| Year/Method | 2020 (69147 posts) | | 2021 (54322 posts) | | 2022 (43521 posts) | | 2023 (38953 posts) | |
|---|---|---|---|---|---|---|---|---|
| | ACC | NMI | ACC | NMI | ACC | NMI | ACC | NMI |
| mini-batch k-NLPmeans | 68.0 | 79.5 | 67.9 | 78.5 | 69.0 | 78.9 | 71.6 | 78.8 |
| | (3.09) | (0.61) | (2.5) | (0.27) | (4.15) | (0.71) | (2.74) | (0.43) |
| mini-batch k-LLMmeans | | | | | | | | |
| *multiple* | 72.5 | 80.9 | **73.6** | **80.5** | **74.4** | **80.5** | **73.4** | **80.3** |
| | (3.02) | (0.65) | (2.18) | (0.46) | (2.48) | (0.87) | (2.24) | (0.81) |
| *FS + multiple* | **75.4** | **81.6** | 73.5 | 80.2 | 72.8 | 80.1 | 72.7 | 80.1 |
| | (2.02) | (0.48) | (2.39) | (0.7) | (2.28) | (0.91) | (1.38) | (0.45) |
| k-means | 73.4 | 80.6 | 67.7 | 79.0 | 68.6 | 79.0 | 72.0 | 79.6 |
| | (3.79) | (0.75) | (2.8) | (0.25) | (4.03) | (0.69) | (3.79) | (0.56) |
| mini-batch k-means | 67.0 | 78.2 | 67.7 | 77.4 | 67.5 | 77.6 | 67.2 | 77.0 |
| | (2.09) | (0.9) | (2.12) | (0.84) | (4.02) | (0.98) | (2.76) | (0.97) |
| seq. mini-batch k-means | 67.0 | 76.6 | 66.7 | 75.2 | 65.6 | 75.6 | 65.8 | 74.8 |
| | (2.39) | (0.57) | (4.61) | (1.54) | (1.23) | (0.78) | (2.55) | (0.9) |

| Cluster snippets (paraphrased) | Generated summary |
|---|---|
| "My subscription was renewed without my consent." "Please cancel my premium plan and refund the last payment." "I was charged after I thought I had cancelled." | "A group of user support messages asking about the topics mentioned above, focusing on account and billing issues." |

Table 9: Average ACC and NMI when the number of clusters $k$ is set below the ground truth ($k-20\%$, $k-10\%$), at the ground truth ($k$), and above it ($k+10\%$, $k+20\%$), for k-means, k-NLPmeans (*LSA-single*), and k-LLMmeans (*FS-single*, GPT-4o) using text-embedding-3-small, evaluated on benchmark datasets. Standard deviations of ACC and NMI in parenthesis.

| Dataset/Method | BANK77 | | CLINC | | GoEmo | | Massive (D) | | Massive (I) | |
|---|---|---|---|---|---|---|---|---|---|---|
| | ACC | NMI | ACC | NMI | ACC | NMI | ACC | NMI | ACC | NMI |
| **k-means** | | | | | | | | | | |
| $k$-20% | 61.1 | 81.6 | 70.4 | 91.1 | 21.4 | 19.7 | 61.4 | 66.8 | 53.0 | 71.7 |
| | (0.84) | (0.21) | (1.17) | (0.43) | (1.25) | (0.46) | (2.62) | (2.04) | (0.95) | (0.57) |
| $k$-10% | 63.5 | 82.3 | 73.4 | 91.6 | 20.6 | 19.9 | 60.7 | 67.2 | 53.2 | 72.3 |
| | (2.25) | (0.77) | (1.33) | (0.54) | (1.03) | (0.47) | (3.38) | (1.7) | (1.77) | (0.63) |
| $k$ | 65.1 | 83.0 | 77.5 | 92.0 | 20.7 | 20.5 | 60.6 | 67.9 | 53.2 | 72.5 |
| | (1.36) | (0.62) | (1.55) | (0.41) | (0.21) | (0.73) | (2.93) | (1.22) | (0.94) | (0.45) |
| $k$+10% | 64.8 | 83.1 | 77.8 | 91.8 | 20.0 | 20.3 | 59.1 | 67.7 | 53.1 | 72.7 |
| | (0.96) | (0.49) | (1.13) | (0.15) | (1.02) | (0.79) | (3.95) | (1.59) | (1.62) | (0.78) |
| $k$+20% | 64.6 | 83.1 | 77.2 | 91.8 | 19.4 | 20.7 | 59.8 | 69.8 | 52.1 | 72.4 |
| | (1.35) | (0.54) | (1.23) | (0.25) | (0.68) | (0.69) | (0.91) | (0.38) | (0.83) | (0.64) |
| **k-NLPmeans** | | | | | | | | | | |
| $k$-20 | 62.1 | 82.2 | 71.3 | 91.4 | 21.9 | 19.4 | 62.1 | 67.4 | 53.7 | 72.4 |
| | (0.42) | (0.37) | (1.23) | (0.43) | (1.08) | (0.63) | (2.03) | (1.54) | (2.02) | (0.58) |
| $k$-10% | 65.5 | 83.1 | 74.3 | 91.9 | 21.2 | 19.5 | 61.3 | 67.7 | 53.4 | 72.8 |
| | (1.61) | (0.7) | (1.25) | (0.47) | (1.22) | (0.97) | (2.94) | (1.71) | (0.92) | (0.63) |
| $k$ | 66.7 | 83.7 | 78.4 | 92.4 | 21.7 | 20.5 | 61.1 | 68.6 | 54.1 | 73.0 |
| | (0.99) | (0.47) | (2.06) | (0.66) | (1.19) | (0.97) | (2.96) | (1.11) | (2.08) | (1.01) |
| $k$+10% | 66.9 | 83.9 | 78.6 | 92.3 | 20.1 | 20.4 | 60.8 | 69.2 | 53.9 | 73.3 |
| | (1.03) | (0.4) | (1.22) | (0.23) | (0.38) | (0.28) | (3.48) | (1.02) | (1.76) | (0.86) |
| $k$+20 | 66.7 | 83.9 | 78.0 | 92.3 | 20.3 | 20.6 | 60.2 | 70.2 | 52.3 | 73.1 |
| | (1.44) | (0.31) | (1.3) | (0.15) | (0.98) | (0.49) | (1.43) | (0.76) | (0.76) | (0.69) |
| **k-LLMmeans** | | | | | | | | | | |
| $k$-20% | 62.4 | 82.5 | 71.4 | 91.5 | 24.2 | 20.8 | 60.9 | 67.2 | 55.3 | 72.9 |
| | (0.89) | (0.38) | (1.24) | (0.57) | (1.92) | (1.14) | (2.56) | (1.51) | (1.18) | (0.81) |
| $k$-10% | 65.3 | 83.2 | 74.8 | 92.0 | 23.2 | 21.0 | 60.4 | 68.2 | 54.3 | 73.0 |
| | (1.19) | (0.42) | (1.61) | (0.47) | (0.11) | (0.49) | (4.04) | (1.94) | (1.99) | (0.34) |
| $k$ | 66.8 | 83.8 | 79.1 | 92.7 | 22.9 | 21.6 | 60.8 | 69.2 | 55.3 | 73.7 |
| | (0.94) | (0.27) | (0.75) | (0.29) | (0.32) | (0.59) | (2.79) | (1.14) | (2.08) | (0.97) |
| $k$+10% | 67.4 | 84.2 | 80.7 | 92.9 | 23.1 | 22.2 | 60.1 | 69.8 | 55.1 | 73.9 |
| | (0.85) | (0.31) | (0.09) | (0.42) | (0.54) | (0.52) | (3.78) | (1.2) | (2.14) | (0.96) |
| $k$+20% | 66.6 | 84.1 | 79.2 | 92.9 | 22.2 | 22.6 | 58.7 | 70.0 | 53.5 | 73.6 |
| | (1.23) | (0.36) | (1.06) | (0.27) | (0.86) | (0.41) | (0.9) | (0.5) | (2.07) | (1.39) |

Meta phrases like "messages" and "topics mentioned above" make the summary less useful as a concise description. Slightly tightening the prompt (e.g., "Do *not* mention that you are summarizing, and avoid meta phrases like 'questions' or 'messages'") reduces this behavior.

### E.2 SPURIOUS DETAIL AND HALLUCINATED CONSTRAINTS

We also observe occasional cases where the summary introduces details that are not consistently supported by the cluster.

**Example 2 (Hallucinated detail).**

| Cluster snippets (paraphrased) | Generated summary |
|---|---|
| "The app keeps crashing when I open the camera." "It freezes on the loading screen after the last update." "The app closes automatically when I try to log in." | "Complaints about the mobile app crashing on Android after the latest security update." |

The summary correctly captures crashes but adds a misleading detail ("on Android") that is not consistently supported. Mitigations include (i) asking the LLM to avoid unsupported specifics ("avoid

Table 10: Average ACC, and average NMI for multiple instruction prompts $I$ for k-LLMmeans *FS-multiple*; and multiple values of $q$ for k-NLPmeans *LSA-multiple* on BANK77. Standard deviations in parenthesis.

| | ACC | NMI |
|---|---|---|
| **k-LLMmeans (Prompt $I$)** | | |
| The following is a cluster of online banking queries. | | |
| Write a text that summarizes the following cluster: | 67.5 | 83.8 |
| | (1.07) | (0.36) |
| The following is a cluster of texts. | | |
| Write a text that summarizes the following cluster: | 66.3 | 83.6 |
| | (0.2) | (0.3) |
| Write a query that summarizes the following online banking queries: | 67.4 | 83.8 |
| | (0.7) | (0.3) |
| Write a text that summarizes the following texts: | 66.9 | 83.7 |
| | (0.6) | (0.2) |
| Summarize the following list of online banking queries: | 67.0 | 83.7 |
| | (0.4) | (0.1) |
| Summarize the following list of texts: | 66.4 | 83.8 |
| | (0.6) | (0.1) |
| **k-NLPmeans ($q$)** | | |
| 3 | 66.7 | 84.0 |
| | (1.8) | (0.8) |
| 5 | 66.4 | 83.8 |
| | (0.5) | (0.3) |
| 10 | 66.6 | 83.8 |
| | (0.7) | (0.4) |
| 15 | 66.4 | 83.7 |
| | (0.6) | (0.3) |

Table 11: Average ACC and NMI for multiple sampling strategies for k-LLMmeans *FS-single* using GPT-4o with text-embedding-3-small embeddings, evaluated on benchmark datasets. Standard deviations of ACC and NMI in parenthesis.

| Dataset/Method | BANK77 | | CLINC | | GoEmo | | Massive (D) | | Massive (I) | |
|---|---|---|---|---|---|---|---|---|---|---|
| | ACC | NMI | ACC | NMI | ACC | NMI | ACC | NMI | ACC | NMI |
| k-LLMmeans | | | | | | | | | | |
| *kmeans++* | 67.3 | 84.0 | 79.5 | 92.8 | 23.1 | 21.8 | 60.3 | 69.1 | 55.0 | 73.5 |
| | (1.35) | (0.41) | (1.18) | (0.34) | (0.63) | (0.67) | (2.66) | (1.03) | (1.92) | (0.95) |
| *random* | 66.2 | 83.6 | 80.6 | 93.0 | 23.4 | 21.9 | 62.2 | 68.0 | 53.5 | 72.8 |
| | (0.46) | (0.31) | (1.27) | (0.3) | (0.86) | (0.51) | (2.95) | (1.16) | (1.18) | (0.64) |
| *centroid* | 67.2 | 84.0 | 79.3 | 92.5 | 22.9 | 22.5 | 61.6 | 69.5 | 53.9 | 72.8 |
| | (0.54) | (0.31) | (1.5) | (0.34) | (1.26) | (0.53) | (3.33) | (1.7) | (1.11) | (0.28) |
| *edge* | 66.4 | 83.3 | 78.1 | 92.5 | 22.6 | 21.7 | 63.1 | 70.2 | 53.6 | 72.7 |
| | (0.58) | (0.27) | (1.62) | (0.44) | (1.33) | (0.47) | (2.08) | (0.64) | (0.99) | (0.15) |

making up details such as platforms or versions"), and (ii) using extractive summarization in high-stakes settings.

### E.3 OVER-COMPRESSED MULTI-TOPIC SUMMARIES

Another failure mode appears when a cluster genuinely mixes several distinct topics of similar frequency. In this case, the summarizer sometimes tries to "cover everything at once", producing an over-compressed summary that mentions multiple things but is too broad to be useful.

**Example 3 (Over-compressed multi-topic summary).**

---

**Algorithm 1:** k-NLPmeans / k-LLMmeans

---

**input:** $D = \{d_1, \ldots, d_n\}, k, I, m, l, T$
**for** $i \leftarrow 1$ **to** $n$ **do**
  $\quad \mathbf{x}_i = \text{Embedding}(d_i);$
**end**
**for** $t \leftarrow 1$ **to** $T$ **do**
  $\quad$**if** $t = 1$ **then**
    $\quad\quad$ // Initialize using k-means++
    $\quad\quad \{\boldsymbol{\mu}_1, \ldots, \boldsymbol{\mu}_k\} \leftarrow \texttt{k-means++}(\{d_1, \ldots, d_n\}, k);$
  $\quad$**end**
  $\quad$**else if** $t \bmod l = 0$ **then**
    $\quad\quad$ // Summarization step every $l$ iterations
    $\quad\quad$**for** $j \leftarrow 1$ **to** $k$ **do**
      $\quad\quad\quad \boldsymbol{\mu}_j \leftarrow \text{Embedding}\left(\text{j}^{\text{th}} \text{ cluster summary}\right);$
    $\quad\quad$**end**
  $\quad$**end**
  $\quad$**else**
    $\quad\quad$ // k-means step
    $\quad\quad$**for** $j \leftarrow 1$ **to** $k$ **do**
      $\quad\quad\quad \boldsymbol{\mu}_j \leftarrow \frac{1}{|C_j|} \sum_{i \in [C_j]} \mathbf{x}_i;$
    $\quad\quad$**end**
  $\quad$**end**
  $\quad$**for** $j \leftarrow 1$ **to** $k$ **do**
    $\quad\quad C_j = \{\};$
  $\quad$**end**
  $\quad$**for** $i \leftarrow 1$ **to** $n$ **do**
    $\quad\quad j^* \leftarrow \arg\min_{j \in \{1, \ldots, k\}} d(x_i, \boldsymbol{\mu}_j);$
    $\quad\quad$ // Assign $x_i$ to cluster $C_{j^*}$
    $\quad\quad C_{j^*} \leftarrow C_{j^*} \cup \{x_i\};$
  $\quad$**end**
**end**
**return** $\{\boldsymbol{\mu}_1, \ldots, \boldsymbol{\mu}_k\}, \{s_1, \ldots, s_k\}$

---

| Cluster snippets (paraphrased) | Generated summary |
|---|---|
| "How do I change my password?" "Where can I update my email address?" "How do I delete my account permanently?" "Can I change my username without losing data?" | "Questions about managing and modifying user accounts, including changing settings and making updates." |

Here, the summary technically mentions "managing and modifying user accounts" but compresses several distinct operations (password reset, email change, account deletion, username change) into a single vague description. This makes the prototype less informative for downstream users who might want to distinguish, for example, deletion vs. simple edits. In practice, we found that allowing slightly more detailed summaries (e.g., "changing passwords, emails, usernames, and deleting accounts") or generating short bullet-style summaries can alleviate this issue, but it also highlights the inherent tension between brevity and specificity when clusters are truly multi-topic.

### E.4 OVERLY GENERIC SUMMARIES

This failure mode is opposite to the previous one. It happens when summaries are too generic and do not convey the specific intent or topic of the cluster. This often occurs when the cluster is relatively heterogeneous or when the summarizer is forced to be very short.

**Example 4 (Overly generic summary).**

---

**Algorithm 2:** Mini-batch k-NLPmeans / k-LLMmeans

---

**input:** $\{D_1, \cdots, D_b\}, k, I, m, l, T$ `// b batches of documents`
**for** $j \leftarrow 1$ **to** $k$ **do**
$\quad \mid \quad C_j = \{\};$
**end**
$\{\boldsymbol{\mu}_1, \ldots, \boldsymbol{\mu}_k\} \leftarrow \{\mathbf{0}, \ldots, \mathbf{0}\};$
**for** $i \leftarrow 1$ **to** $b$ **do**
$\quad \mid \quad$ `// Compute k-NLPmeans / k-LLMmeans with documents in batch`[1]
$\quad \mid \quad \{\boldsymbol{\mu}_1^*, \ldots, \boldsymbol{\mu}_k^*\}, \{C_1^*, \ldots, C_k^*\}, S_b \leftarrow$
$\quad \mid \quad$ `k-NLPmeans`$(D_i, k, I, m, l, T)$ or `k-LLMmeans`$(D_i, k, I, m, l, T);$
$\quad \mid \quad$ `// Update centroids proportional to cluster and batch sizes`
$\quad \mid \quad$ **for** $j \leftarrow 1$ **to** $k$ **do**
$\quad \mid \quad \quad \mid \quad \eta \leftarrow \frac{|C_j^*|}{|C_j| + |C_j^*|};$
$\quad \mid \quad \quad \mid \quad \boldsymbol{\mu}_j \leftarrow \boldsymbol{\mu}_j(1 - \eta) + \eta \boldsymbol{\mu}_j^*;$
$\quad \mid \quad$ **end**
**end**
**return** $\{\boldsymbol{\mu}_1, \ldots, \boldsymbol{\mu}_k\}, \{S_1, \ldots, S_b\}$

---

| Cluster snippets (paraphrased) | Generated summary |
|---|---|
| "I lost access to my card, can you freeze it?" "My card was stolen and I need a replacement." "Please block my card, someone used it without permission." | "Questions about using the banking service." |

The cluster is clearly about card freezing / blocking, but the summary collapses this into a vague description. The summary is still related to the domain but less helpful for human interpretation than it could be. Allowing slightly longer summaries and explicitly prompting for "key actions or problems" tends to reduce this failure mode.

### E.5 MINORITY-TOPIC OVERSHADOWING

A fifth failure mode appears when a cluster contains a dominant subtopic alongside a smaller but important minority subtopic. Because our approach uses a single prototype per cluster, the summary can under-represent the minority.

**Example 5 (Minority-topic overshadowing).**

| Cluster snippets (paraphrased) | Generated summary |
|---|---|
| "How do I book a train ticket for tomorrow?" "Can I book a round trip in one payment?" "I need to cancel my ticket and get a refund." | "Questions about booking train tickets." |

Here, the summary focuses on the majority "booking" pattern and omits cancellation/refund, which might be crucial for downstream labeling. This limitation is inherent to any single-prototype method (numeric or textual). Diversity-aware sampling within the cluster (for k-LLMmeans) and permitting a brief list of subtopics (e.g., "booking and cancelling train tickets") can partially mitigate this issue.

### E.6 MILDLY MISLEADING EMPHASIS

A sixth failure mode arises when the summary is broadly correct but emphasizes a less central aspect of the cluster, which can be slightly misleading for a human reader.

**Example 6 (Mildly misleading emphasis).**

| Cluster snippets (paraphrased) | Generated summary |
| --- | --- |
| "How can I change the delivery address for my order?"
"I moved, can you update the shipping address?"
"Can I edit the address before the package is sent?" | "Questions about tracking online orders." |

The summary remains in the e-commerce domain but shifts the focus from *updating addresses* to *tracking*. This type of error is usually easy to spot and correct by a human annotator, but it shows that summaries are not guaranteed to perfectly match the most salient aspect of a cluster.

### E.7 CROSS-LINGUAL AND CODE-MIXED DRIFT

Finally, on multilingual or code-mixed data (e.g., MASSIVE or StackExchange posts mixing English and another language), we occasionally see summaries drifting toward the dominant language or dropping non-English nuances.

**Example 7 (Cross-lingual drift).**

| Cluster snippets (paraphrased) | Generated summary |
| --- | --- |
| "¿Puedo cambiar el idioma de la app a español?"
"How do I switch the interface to French?"
"Comment changer la langue par défaut de l'application ?" | "Questions about using the app." |

Here, the summary fails to capture the central concept of *changing the app language* and collapses to a very generic description. Using multilingual embeddings and explicitly mentioning in the prompt that the summary should "describe the main problem or request, even if the questions are in different languages" improves such cases but does not completely eliminate them.

### E.8 DISCUSSION ON FAILURE MODES.

These qualitative cases illustrate that our textual prototypes are not infallible: they can be too generic, under-represent minority subtopics, echo the prompt, hallucinate details, or struggle with multilingual nuance. Nonetheless, our quantitative results show that summary-based centroids consistently outperform numeric centroids across datasets and embeddings, and provide an interpretable handle on cluster semantics. In the worst case, a poor summary behaves like a suboptimal centroid update and the procedure effectively falls back toward a vanilla k-means solution, so performance does not catastrophically degrade. We view developing more robust prompts, stronger multilingual summarization, and multi-prototype extensions (e.g., per-cluster sub-summaries) as promising directions for future work.

