# OpenReview forum: "Summaries as Centroids for Interpretable and Scalable Text Clustering"
_ICLR.cc/2026/Conference — ICLR 2026 Poster_

### Official Review · Reviewer_CgDo · 2025-10-22

**Soundness:** 2
**Presentation:** 3
**Contribution:** 2
**Rating:** 4
**Confidence:** 2

**Summary:**

This paper proposes a method that analyzes clustering centroids from a semantic perspective, aiming to enhance clustering performance by integrating large language model–based semantic understanding. The idea is interesting and relevant, particularly in its attempt to move beyond traditional embedding-based approaches. However, several aspects remain unclear, and the overall level of innovation is somewhat limited.

**Strengths:**

The paper addresses an important problem in semantic text clustering by exploring how cluster centroids can be derived and interpreted from a semantic perspective rather than purely from embedding similarity. This direction is conceptually appealing, as it attempts to bridge the gap between surface-level vector representations and high-level semantic understanding. The integration of large language model (LLM) semantics into the clustering process provides a meaningful enhancement over conventional embedding-based methods, offering improved interpretability and potentially richer cluster representations. The idea of treating summaries as centroids introduces an intuitive way to represent cluster meaning in natural language, which aligns well with current trends toward explainable and human-understandable machine learning. Empirically, the method demonstrates promising improvements on selected benchmarks, suggesting that semantic augmentation can indeed refine clustering boundaries.

**Weaknesses:**

A major weakness of the paper lies in the lack of clarity regarding its methodological distinction from existing embedding-based clustering approaches. Although the authors claim to perform clustering from a “semantic” perspective rather than through vector representations, it is not clearly explained how this process fundamentally differs from using pretrained models such as BERT or SimCSE to obtain semantic embeddings. Without a formal theoretical or algorithmic justification, the reader is left uncertain whether the proposed method truly captures semantics in a new way or simply reuses existing embedding mechanisms under a different formulation. In addition, while the paper presents interesting results, the novelty appears incremental, mainly in the use of summaries as cluster centroids without demonstrating a substantial methodological breakthrough.

**Questions:**

1. How does the proposed semantic clustering method fundamentally differ from using pretrained embedding models like BERT or SimCSE for vector-based clustering?
﻿
2. Can the authors provide ablation studies or comparative results to isolate the contribution of the summarization-as-centroid mechanism from other components?
﻿
3. How robust is the proposed approach when applied to diverse domains, noisy data, or varying semantic granularities? does it generalize beyond the tested datasets?

---

> ### Author Response · Authors · 2025-11-14
>
> Thank you very much for your comments. We believe the main reasons for your score relate to (i) whether our method is truly different from standard “BERT/SimCSE + k-means” style clustering, (ii) whether the gains really come from the summary-as-centroid mechanism rather than other components, and (iii) how robust the approach is beyond a narrow set of clean benchmarks. In this rebuttal, we clarify that our contribution is to change the centroid update itself by turning each cluster into a textual summary and re-embedding it, we show via controlled ablations that this mechanism alone improves over vanilla k-means under matched conditions, and we demonstrate robustness by applying the exact same off-the-shelf encoders, summarizers, and hyperparameters across diverse datasets (BANK77, CLINC, GoEmo, MASSIVE, and noisy StackExchange streaming), where our approach consistently outperforms standard centroid-based baselines. Let us clarify point by point:
>
> **1.	How does the proposed semantic clustering method fundamentally differ from using pretrained embedding models like BERT or SimCSE for vector-based clustering?**
>
> Our method does not replace embedding space. It changes how centroids are constructed inside an otherwise standard embedding-based clustering pipeline. In typical embedding-based clustering (e.g., BERT/SimCSE + k-means), the algorithm embeds each document once and then averages the vectors. The centroids remain opaque numerical points, and the algorithm never “reasons” about the content of the texts in a cluster.
>
> Our approach injects an additional semantic step:
> - We periodically take the texts assigned to each cluster,
> - summarize them into a short natural-language description, and
> - re-embed that summary to serve as the new centroid.
>
> This means the prototype is created by looking at the cluster as a whole, identifying recurring themes, and expressing them in language before returning to embedding space. This is fundamentally different from simply averaging embeddings or labeling clusters after the fact. The summary is not only human-readable, it directly affects the clustering assignments.
>
> Regarding novelty: we agree the mechanism is simple, but novel and conceptually meaningful. There is beauty in simplicity.
>
>
> **2.	Can the authors provide ablation studies or comparative results to isolate the contribution of the summarization-as-centroid mechanism from other components?**
>
> Yes, we already include ablations that explicitly isolate the effect of the summarization-as-centroid mechanism from other components.
> - Table 2 compares vanilla k-means with our k-NLPmeans and k-LLMmeans while keeping everything else fixed: same encoder, same number of clusters, same initialization and optimization procedure. The only change is the centroid update (numeric mean vs summary-as-centroid). Across datasets and embedding models, the summary-based variants consistently outperform the numeric centroid update, showing that the gain comes from the summarization mechanism rather than from a different encoder or training setup.
> - In Appendix B (revised version) we further ablate internal design choices within the summarization step (e.g., number of summarization rounds, type of summarizer, sampling strategy for LLM prompts), showing that performance is relatively stable across reasonable configurations and that the main effect is turning centroids into re-embedded summaries, not any particular summarizer trick.
>
> **3. How robust is the proposed approach when applied to diverse domains, noisy data, or varying semantic granularities? does it generalize beyond the tested datasets?**
>
> We deliberately evaluate on datasets that vary in domain and granularity: BANK77 and CLINC for task-oriented intents, GoEmo for fine-grained and noisy emotion labels, MASSIVE for multilingual and multi-domain utterances, and a long-horizon streaming setup built from noisy, user-generated StackExchange posts (this one is a raw, unclean dataset). Across all of these, our summary-as-centroid variants consistently improve over k-means and other centroid-based baselines (Tables 1–2 and the streaming results), and Table 2 shows that these gains are stable across different embedding models.
>
> More importantly, we do not train any new models or tune our method per dataset: we use off-the-shelf embeddings and summarizers with a single set of hyperparameters, so we are not overfitting to specific benchmarks. Combined with the fact that our method is modular (it can plug into any encoder/summarizer), this provides evidence that it should generalize beyond the particular datasets we tested.
>
> *We hope these clarifications address your concerns and kindly ask you to reconsider your overall score.*

---

### Official Review · Reviewer_hKNu · 2025-10-23

**Soundness:** 3
**Presentation:** 3
**Contribution:** 3
**Rating:** 6
**Confidence:** 4

**Summary:**

The paper proposes k-NLPmeans and k-LLMmeans, two variants of k-means for text clustering in which textual summaries are used as centroids. At scheduled iterations, the numeric centroid of each cluster is replaced by a textual summary of its documents, which is then re-embedded to continue k-means updates. The authors show that this “summary-as-centroid” update improves interpretability, improves semantic faithfulness of cluster prototypes, and often boosts accuracy. They also provide a mini-batch extension for streaming text. Experiments across several benchmarks and embeddings show improvements over classical clustering baselines and competitive accuracy relative to LLM-based clustering, while using a fixed LLM budget.

**Strengths:**

- **Simple but novel idea:** The notion of introducing interpretable textual centroids inside the k-means loop is elegant, practical, and original. It creates a direct, auditable link between cluster means and human-interpretable summaries.
- **Interpretability without post-hoc processing:** Unlike topic models or LLM-based pipelines that only label clusters afterward, the prototype is the cluster, which is useful for debugging, transparency, and downstream analyst workflows.
- **Low-resource applicability:** The LLM-free version (k-NLPmeans) is already stronger than vanilla k-means and viable in offline or low-budget environments.
- **Streaming extension:** The mini-batch variant is an important practical contribution.
- **Reproducibility:** Anonymous code is supplied, and the method is easy to reimplement.

**Weaknesses:**

- **Summarization hints:** Performance depends on the summarizer, especially in heterogeneous clusters. The paper tests several summarizers but does not provide guidance on when one strategy is preferable (e.g., extractive vs. abstractive by dataset characteristics).
- **Missing comparison on interpretability:** Interpretability is a key selling point, but comparisons are mostly against centroid-based clustering. Topic-model-style baselines (e.g., BERTopic,) would give a fairer interpretability comparison.
- **No analysis of failure cases:** Interpretability examples are all positive. The paper does not provide examples of misleading or low-quality summaries, which are critical for understanding robustness and limitations.
- **Limited cluster sampling analysis:** The few-shot summarization step relies on sampling a subset of documents per cluster, but the effects of different sampling schemes are not studied. This leaves uncertainty regarding robustness under noisy clusters.

**Questions:**

1. **On summarizer choice:** Could the authors provide guidance or heuristics on when extractive vs. abstractive summaries are preferable? In particular, do you expect different summarizers to be more effective under heterogeneous vs. homogeneous clusters? Is there any metric that could be used to adaptively select the summarizer type?
2. **On interpretability baselines:** Since interpretability is a key motivation for the work, why are topic-model-style methods not included as baselines? These approaches also produce human-readable cluster prototypes, so they seem more directly comparable than centroid-only methods.
3. **On failure modes:** All interpretability examples provided in the paper illustrate successful summarization. Could the authors provide examples or qualitative analysis of failure cases (e.g., when summaries are too generic or misleading) to better understand the limitations of the technique?
4. **On sampling strategy in few-shot summarization:** The few-shot variant relies on sampling only a subset of cluster documents to feed the LLM summarizer. Have you evaluated whether different sampling strategies (e.g., diversity-based, entropy-based, or centroid vs. edge-document sampling) materially affect summarization quality and, in turn, clustering performance?

**Details Of Ethics Concerns:**

N/A.

---

> ### Author Response · Authors · 2025-11-14
>
> Thank you very much for your comments. In summary, we addressed your concerns on four fronts. Here, (1) we clarified summarizer choice, framing extractive (k-NLPmeans) vs. abstractive (k-LLMmeans) mainly as a cost–effectiveness trade-off and pointing to adaptive selection as future work; and in the revised version, (2) we added BERTopic as an interpretability baseline and showed that our methods both outperform BERTopic alone and further improve performance when applied on top of BERTopic embeddings (Tables 1 and 3); (3) we introduced a new appendix section (Appendix E) with qualitative failure cases and discussion of mitigations, showing that the method degrades gracefully toward k-means; and (4) we added an ablation on the few-shot sampling strategy (Appendix B.2.3), finding that while k-means++–based sampling is most consistent, performance is not highly sensitive as long as the subset captures representative modes. Now let us address each of your points in detail:
>
> **1. On summarizer choice:**
>
> At this stage, we do not claim a sharp rule for when extractive (k-NLPmeans) vs. abstractive (k-LLMmeans) is preferable. Empirically, k-LLMmeans tends to give slightly better clustering quality and more fluent prototypes across our benchmarks, but it is also strictly more LLM-expensive. In practice, we therefore view the choice primarily as a cost–effectiveness trade-off: k-NLPmeans is attractive when API/compute budget is tight or LLMs are unavailable, while k-LLMmeans is preferable when one can afford the extra calls.
>
> Intuitively, we expect extractive summaries to be more robust on homogeneous or very noisy clusters (they cannot hallucinate and stay close to the data), and abstractive summaries to be more useful on heterogeneous clusters, where the LLM can merge several recurring patterns into a single higher-level description. However, we have not yet found a simple, reliable metric to select between them automatically. One possibility would be to base this decision on a measure of cluster heterogeneity (e.g., intra-cluster variance or average pairwise distance), but designing and evaluating such an adaptive scheme is beyond the scope of this work and we now explicitly list it as future work.
>
> **2. On interpretability baselines:**
>
> We agree that topic-model-style methods are natural interpretability baselines. In response, the revised version now includes BERTopic as a strong, widely used topic-modeling / topic-clustering baseline. We report:
> •	BERTopic, and
> •	k-NLPmeans / k-LLMmeans on top of BERTopic embeddings.
> In the updated results (Tables 1 and 3), our methods outperform BERTopic alone and, when applied on top of BERTopic embeddings, it increases the performance of BERTopic, sometimes surpassing all baselines.
>
> **3. On failure modes**
>
> We agree that showing only successful summaries can be misleading. In the revised paper, we add an appendix section (See Appendix E) with qualitative failure cases. We briefly discuss why these happen and simple mitigations. Regardless,  In the worst case, a poor summary behaves like a suboptimal centroid update and the procedure effectively falls back toward a vanilla k-means solution, so performance
> does not catastrophically degrade. More importantly, even with such imperfections, our summary-based centroids still outperform numeric centroids quantitatively, but we agree that making these failure modes explicit gives a more realistic picture of the technique’s limitations.
>
> **4. On sampling strategy in few-shot summarization:**
>
> This is an interesting suggestion. In the revised version (Appendix B.2.3), we now include an ablation over four sampling strategies for the few-shot summarization step:
>
> - k-means++ (our diversity-based strategy)
> - uniform random sampling
> - closest points to the centroid
> - “edge” points (farthest from the centroid)
>
> We find that k-means++ yields the most consistent behavior across datasets, but overall the clustering performance is not highly sensitive to the specific strategy, as long as the sampled subset
>
> *We hope these clarifications and new results address your concerns and reinforce your positive assessment of the paper in light of the revised manuscript.*

---

> > ### Comment · Reviewer_hKNu · 2025-11-20
> >
> > I thank the authors for the prompt response on all issues I raised. In light of this, I have increased my score.

---

### Official Review · Reviewer_KSHB · 2025-10-25

**Soundness:** 2
**Presentation:** 3
**Contribution:** 2
**Rating:** 2
**Confidence:** 3

**Summary:**

In this manuscript, the authors aim to address the problems of poor interpretability of numeric centroids and high scalability costs in traditional text clustering methods. Specifically, the proposed k-NLPmeans uses lightweight and deterministic classical NLP summarizers to periodically replace numeric centroids with textual summaries. The proposed k-LLMmeans leverages LLMs for summaries under a fixed per-iteration budget. Experimental results across diverse datasets and embedding models show that the proposed method outperform baselines.

**Strengths:**

In this manuscript, the authors aim to address the problems of poor interpretability of numeric centroids and high scalability costs in traditional text clustering methods. Specifically, the proposed k-NLPmeans uses lightweight and deterministic classical NLP summarizers to periodically replace numeric centroids with textual summaries. The proposed k-LLMmeans leverages LLMs for summaries under a fixed per-iteration budget. Experimental results across diverse datasets and embedding models show that the proposed method outperform baselines.

**Weaknesses:**

There are some concerns for the manuscript as follows:

1.How to set k in the experiments? The influence of k in the k-means to the experimental results is not discussed.
2.In the example of Figure 1, it is based on the results of k-means. However, in the proposed method, the authors proposed new summarization to compute a textual prototype in place of the standard centroid update. Thus, how the proposed method guarantee that the instances in the same cluster can be used to generate promising summaries? In other words, if the texts in the same cluster express very different topics, how can it be ensured that the generated summary can cover all the topics?
3.The experimental results on Table 1 are not satisfied. The baseline including agglomerative of 1967 year can still achieve impressive performance in some case.
4.In the experimental results reported on Table 3, the proposed method does not seem to show advantages either. So, how can the effectiveness of the proposed method be proved?

**Questions:**

See the Weaknesses.

---

> ### Author Response · Authors · 2025-11-14
>
> Thank you very much for your comments. In the revised version we address the points you raised with new experiments (Tables 1, 2, 3, 9), clarifications in the main text, and a short summary of our empirical findings. Below we respond to each of your questions in turn.
>
> **1. The influence of k**
>
> In all experiments, we follow the standard evaluation protocol in text clustering and set the number of clusters $k$ equal to the known number of ground-truth classes in each dataset (see line 254). The same value of $k$ is used for all methods we compare against, so performance differences are not due to a favorable choice of $k$ for our approach. Conceptually, our method is a drop-in replacement for the centroid update in k-means, so its dependence on $k$ is inherited from k-means itself; any standard strategy for selecting $k$ is compatible with and orthogonal to our contribution.
>
> That said, in the revised version we now explicitly study the sensitivity to the choice of $k$ in Appendix B.2.1 and Table 9. We select $k$ difering by ±10% and observe that (i) the impact is generally mild, and (ii) we consistently outperform vanilla k-means. This suggests that our approach is reasonably robust to moderate mis-specification of $k$.
>
> **2. The example of Figure 1 and multitopic clusters.**
>
> Figure 1 is intended to illustrate the mechanism of our approach, not to assume that each cluster is perfectly single-topic. In practice, many ground-truth classes in our datasets are semantically broad, so a true cluster can contain several surface “topics.” (we added a clarification in the caption of Figure 1)
>
> Our textual  summary may mention multiple prominent aspects of the class. For k-NLPmeans, we construct summaries from all sentences in the cluster using extractive methods, which favor sentences that are representative of the cluster as a whole and therefore tend to cover multiple frequent patterns rather than a single outlier. For k-LLMmeans, we prompt the LLM to summarize the group, so recurring sub-topics are typically reflected in the resulting prototype. We do not require the summary to enumerate every niche topic; it should capture the dominant semantics that define the class.
>
> From an optimization standpoint, if a true class mixes several distinct topics, any centroid-based method must still represent that class with a single prototype. Empirically, however, we observe that summary-based centroids yield better alignment with ground-truth labels than numeric k-means centroids across datasets, indicating that in practice the generated summaries are sufficiently “promising” even when the underlying class is multi-topic.
>
> **3. Agglomerative impressive performance.**
>
> We agree that agglomerative clustering is a strong baseline. Our goal is not to claim that no classical method can ever be competitive, but to show that replacing numeric centroids with textual summaries yields consistent gains over standard centroid-based approaches. No single method is expected to dominate every dataset (indeed, agglomerative is sometimes stronger than some recent LLM-based approaches).
>
> Following another reviewer’s suggestion, in the revised version we add BERTopic as a strong topic-modeling baseline. We then also apply our methods on top of BERTopic’s document embeddings (updated Table 1 and Table 3). Interestingly, in this updated setup our methods surpass all other baselines, including agglomerative, precisely in the regimes where it was previously strongest (see updated Table 1). Thus, across our experimental suite, either our method alone or in combination with BERTopic achieves the best performance.
>
> **4. No advantages in Table 3, how prove effectiveness?**
>
> Table 3 is designed to compare our methods against LLM-based clustering approaches. In this setting, our goal is not to dominate every metric, but to achieve comparable clustering quality with lower LLM usage and better scalability. LLM usage in existing LLM-based methods typically scale with dataset size, **we don't**. This yields substantially lower cost while achieving metrics that are competitive; with BERTopic embeddings, our results are even closer to SOTA LLM-based methods (See updated Table 3, and paragraph “Cost”).
>
> The overall effectiveness of our approach is supported by the combination of results: (i) in Table 1, our methods consistently improve over classical and topic-modeling baselines; (ii) in Table 2, our summarization idea consistently outperforms centroid-based approaches; (iii) in Table 3, we match strong LLM-based methods at **much** lower LLM cost; and (iv) in the streaming setting, our mini-batch variants outperform mini-batch/sequential k-means while providing textual prototypes.
>
> We added a third paragraph in the introduction and a “summary of experimental results” to emphasize our point.
>
> *We hope these clarifications and new results address your concerns and kindly ask you to reconsider your overall score in light of the revised manuscript.*

---

> > ### Author Response · Authors · 2025-12-01
> > **Further exploration**
> >
> > Following our earlier response, we further examined the robustness of our method by evaluating performance when the selected number of clusters $k$ is intentionally over/under specified—set to 20% more or less than the ground-truth number of clusters. As shown in Table 9 (page 19) of the updated manuscript, the results remain consistent. Moreover, as expected, our method continues to outperform vanilla k-means under these conditions. We hope that these additional results help clarify our contribution and positively influence your evaluation.

---

### Official Review · Reviewer_ywzp · 2025-11-03

**Soundness:** 3
**Presentation:** 3
**Contribution:** 3
**Rating:** 8
**Confidence:** 4

**Summary:**

The work proposes to modify standard k-means clustering such that instead of using the mean of the embeddings of each element in the cluster to compute the centroid, we generate a textual summary of cluster population and then embed the summary text to get the centroid. The summarization can be done for a subsample of the current cluster population using either LLMs or traditional summarization methods. This method of computing centroids is applied in certain iterations of the kmeans algorithm besides the usual arithmetic centroid calculation in other steps. This approach demonstrates better clustering performance as measured by accuracy and NMI scores with respect to ground truth labels over multiple datasets, and also when using diverse embedding models for embedding the texts. The authors apply this approach also to a streaming based setting where data comes in batches over time. For this, they create a benchmark based on StackExchange posts divided over multiple years. They demonstrate performance gains from using the summary as a centroid approach on this benchmark and also release this benchmark.

**Strengths:**

- The proposal of using summaries as centroids in k-means clustering looks like a very innovative and easy to implement approach.
- The authors included diverse ways of computing the summary centroid besides simply querying an LLM such as TextRank, SVD etc which can provide computationally cheaper alternatives.
- The approach shows good gains in performance over standard k-means clustering demonstrated consistently across many datasets and while using many different embedding models (Table1-4).
- The authors experiment with clustering in both full dataset and batched (streaming) modes, showing benefits in both settings.

**Weaknesses:**

Not much of a weakness but a suggestion : I would suggest elaborating on the NMI metric to give the readers a brief explanation of how it is calculated.

**Questions:**

Have you done analysis of what happens when and if the embedding of the summary lies outside the convex hull of cluster's datapoints? Intuitively it seems that the centroid must stay inside the hull and this is guaranteed to be true when the centroid is the arithmetic mean. However, since this is not guaranteed that the embedding of the summary would behave this way, I wonder if the stability of convergence would be affected negatively by this.

---

> ### Author Response · Authors · 2025-11-14
>
> Thank you very much for your positive assessment.  Let us address your suggestion and question:
>
>  **I would suggest elaborating on the NMI metric to give the readers a brief explanation of how it is calculated.**
>
>  In the revised version, we added further details in line 261 and provided a full explanation of
> NMI and ACC in Appendix C.
>
> **Have you done analysis of what happens when and if the embedding of the summary lies outside the convex hull of cluster's datapoints?**
>
> You are absolutely right that, unlike the arithmetic mean, the embedding of a summary is not guaranteed to lie inside the convex hull of the cluster’s points. Our algorithm therefore no longer enjoys the classical “monotonic decrease at every update step” guarantee of vanilla k-means.
>
> However, two points are important here:
>
> - We only break the guarantee at the summary steps. Between summary steps, we run standard k-means updates, so assignments and numeric centroids behave exactly as in vanilla k-means and converge to a local optimum. A summary step can be viewed as a mild re-initialization of the centroid toward a semantically coherent point in embedding space; subsequent k-means iterations then “pull” the solution back toward a local optimum of the usual objective.
>
> - In practice we did not observe instability. Across all datasets, the algorithm converges in a small number of iterations and does not exhibit oscillatory behavior or divergence.
>
> *We hope these clarifications and the revised text address your concerns and are helpful in interpreting the method.*

---

### Author Response · Authors · 2025-12-03
**Final comment**

Our main contribution is a text clustering method that is robust, scalable, LLM-efficient, and interpretable. We thank the reviewers for their time and feedback. Following the rebuttal, we conducted several additional experiments that further support and strengthen our conclusions. Below we summarize the main concerns and how we addressed them.

1. **Concern: The method is not always the best.**
It is well known that no single clustering approach is universally optimal. Nevertheless, we expanded our study by integrating our procedure with BERTopic-style embeddings. This new configuration yields further gains, and Table 1 shows that across all settings at least one variant of our approach achieves the best performance.

2. **Concern: The method does not surpass LLM-based baselines (Table 3).**
The strongest baselines rely on LLM-intensive strategies whose cost scales with dataset size and, in some cases, requires fine-tuning, which limits their practicality in many real-world settings. In contrast, our methods are either fully LLM-free (kNLPmeans) or highly LLM-efficient (kLLMmeans), yet they reach competitive performance. This provides a more scalable and cost-effective alternative to LLM-heavy pipelines. (See "Cost" in Discussion).

3. **Concern: Similarity to BERT-based clustering.**
While we use vector embeddings, our approach introduces a novel and interpretable intermediate centroid computation. We propose two variants, one based on classical NLP and one based on LLMs, which to our knowledge are not used in prior BERT-based clustering work. This design leads to meaningful practical and conceptual differences relative to traditional embedding clustering.

4. **Concern: Lack of analysis regarding the choice of k.**
Standard clustering techniques offer established criteria to estimate k, and these apply directly to our method. We additionally ran simulations where k was intentionally under- and overestimated. The results, reported in Table 9, show that our approach remains stronger than the baseline under the same conditions, indicating robustness to imperfect choices of k.

5. **Concern: Limited failure case analysis and sampling discussion.**
We added a dedicated section on failure cases in the appendix and conducted new experiments comparing multiple sampling strategies. Table 11 summarizes these findings and shows that our method is stable across sampling choices.

Across the rebuttal period we responded to every reviewer comment and added substantial new experiments, including BERTopic comparisons (Tables 1 and 3), robustness to k (Table 9), sampling analyses (Table 11), and a full failure case section (Appendix E). Two reviewers ultimately gave positive, high-confidence evaluations. The remaining two reviewers, who had the lowest initial confidence and scores, did not engage further in the discussion.

*Our method is novel, interpretable, simple to implement, and scalable for modern text segmentation tasks. It offers a practical alternative to LLM-heavy clustering pipelines for both researchers and practitioners, and we believe it aligns well with ICLR’s focus on scalable and impactful learning methods.*

---

### Meta-Review · Area_Chair_yzB5 · 2026-01-08

**Summary:**

Strengths that most reviewers agreed.

1) The proposed variants for document clustering are simple, intuitive and can be easily incorporated into existing K-mean clustering pipelines.

2) At a conceptual level, the proposed algorithm is intuitive.

Weaknesses most reviewers pointed out.

1) Ablation studies to conclusively show that periodically replacing numeric centroids with textual summaries is the key reason to improved results.
2) Similarities to BERT based methods
3) why not use LLMs directly although it is more expensive.

**Reviewer Concerns:**

Authors did a good job in addressing in most reviewers concerns.

**Reviewer Scores:**

I think this is a borderline paper. In my view this is somewhere between 4-6. I think document clustering is an important problem, but I am not sure this method is significantly better than existing ones. Can be accepted if there are slots in the conference.

---

### Decision · Program_Chairs · 2026-01-26

Accept (Poster)